# Integrating Symmetry into Differentiable Planning with Steerable Convolutions

**Linfeng Zhao**[*]**, Xupeng Zhu**[†]**, Lingzhi Kong**[†]**, Robin Walters, Lawson L.S. Wong**
Khoury College of Computer Sciences, Northeastern University

## Abstract

In this paper, we study a principled approach on incorporating group symmetry into end-to-end differentiable planning algorithms and explore the benefits of symmetry in planning. To achieve this, we draw inspiration from equivariant convolution networks and model the path planning problem as a set of signals over grids. We demonstrate that value iteration can be treated as a linear equivariant operator, which is effectively a steerable convolution. Building upon Value Iteration Networks (VIN), we propose a new Symmetric Planning (SymPlan) framework that incorporates rotation and reflection symmetry using steerable convolution networks. We evaluate our approach on four tasks: 2D navigation, visual navigation, 2 degrees of freedom (2-DOF) configuration space manipulation, and 2-DOF workspace manipulation. Our experimental results show that our symmetric planning algorithms significantly improve training efficiency and generalization performance compared to non-equivariant baselines, including VINs and GPPN.

## 1 Introduction

Model-based planning algorithms can struggle to find solutions for complex problems, and one solution is to apply planning in a more structured and reduced space (Sutton and Barto, 2018; Li et al., 2006; Ravindran and Barto, 2004; Fox and Long, 2002). When a task exhibits symmetry, this structure can be used to effectively reduce the search space for planning. However, existing planning algorithms often assume perfect knowledge of dynamics and require building equivalence classes, which can be inefficient and limit their applicability to specific tasks (Fox and Long, 1999; 2002; Pochter et al., 2011; Zinkevich and Balch, 2001; Narayanamurthy and Ravindran, 2008).

Figure 1: Symmetry in path planning. Symmetric Planning approach guarantees the solutions are same up to rotations.

In this paper, we study the path-planning problem and its symmetry structure, as shown in Figure 1. Given a map $M$ (top row), the objective is to find optimal actions $A = \texttt{SymPlan}(M)$ (bottom row) to a given position (red dots). If we rotated the map $g.M$ (top right), its solution $g.A$ (shortest path) can also be connected by a rotation with the original solution $A$. Specifically, we say the task has *symmetry* since the solutions $\texttt{SymPlan}(g.M) = g.\texttt{SymPlan}(M)$ are related by a $\circlearrowleft 90°$ rotation. As a more concrete example, the action in the NW corner of $A$ is the same as the action in the SW corner of $g.A$, after also rotating the arrow $\circlearrowleft 90°$. This is an example of symmetry appearing in a specific task, which can be observed *before* solving the task or assuming other domain knowledge. If we can use the rotation (and reflection) symmetry in this task, we effectively reduce the search space by $|C_4| = 4$ (or $|D_4| = 8$) times. Instead, classic planning algorithms like A* would require searching symmetric states (NP-hard) with known dynamics (Pochter et al., 2011).

Recently, symmetry in model-free deep reinforcement learning (RL) has also been studied (Mondal et al., 2020; van der Pol et al., 2020a; Wang et al., 2021). A core benefit of model-free RL

---

[*]Corresponding Author: Linfeng Zhao `zhao.linf@northeastern.edu`.
[†]Second and third authors contributed equally.

that enables great asymptotic performance is its end-to-end differentiability. However, they lack long-horizon planning ability and only effectively handle pixel-level symmetry, such as flipping or rotating image observations and action together. This motivates us to combine the spirit of both: *can we enable end-to-end differentiable planning algorithms to make use of symmetry in environments?*

In this work, we propose a framework called Symmetric Planning (SymPlan) that enables planning with symmetry in an end-to-end differentiable manner while avoiding the explicit construction of equivalence classes for symmetric states. Our framework is motivated by the work in the equivariant network and geometric deep learning community (Bronstein et al., 2021a; Cohen et al., 2020; Kondor and Trivedi, 2018; Cohen and Welling, 2016a;b; Weiler and Cesa, 2021), which views geometric data as signals (or "steerable feature fields") over a base space. For instance, an RGB image is represented as a signal that maps $\mathbb{Z}^2$ to $\mathbb{R}^3$. The theory of equivariant networks enables the injection of symmetry into operations between signals through equivariant operations, such as convolutions. Equivariant networks applied to images do not need to explicitly consider "symmetric pixels" while still ensuring symmetry properties, thus avoiding the need to search symmetric states.

We apply this intuition to the task of path planning, which is both straightforward and general. Specifically, we focus on the 2D grid and demonstrate that value iteration (VI) for 2D path planning is equivariant under translations, rotations, and reflections (which are isometries of $\mathbb{Z}^2$). We further show that VI for path planning is a type of steerable convolution network, as developed in (Cohen and Welling, 2016a). To implement this approach, we use Value Iteration Network (VIN, (Tamar et al., 2016a)) and its variants, since they require only operations between signals. We equip VIN with steerable convolution to create the equivariant steerable version of VIN, named SymVIN, and we use a variant called GPPN (Lee et al., 2018) to build SymGPPN. Both SymPlan methods significantly improve training efficiency and generalization performance on previously unseen random maps, which highlights the advantage of exploiting symmetry from environments for planning. Our contributions include:

- We introduce a framework for incorporating symmetry into path-planning problems on 2D grids, which is directly generalizable to other homogeneous spaces.

- We prove that value iteration for path planning can be treated as a steerable CNN, motivating us to implement SymVIN by replacing the 2D convolution with steerable convolution.

- We show that both SymVIN and a related method, SymGPPN, offer significant improvements in training efficiency and generalization performance for 2D navigation and manipulation tasks.

## 2 RELATED WORK

**Planning with symmetries.** Symmetries are prevalent in various domains and have been used in classical planning algorithms and model checking (Fox and Long, 1999; 2002; Pochter et al., 2011; Shleyfman et al., 2015; Sievers et al., 2015; Sievers; Winterer et al.; Röger et al., 2018; Sievers et al., 2019; Fišer et al., 2019). Invariance of the value function for a Markov Decision Process (MDP) with symmetry has been shown by Zinkevich and Balch (2001), while Narayanamurthy and Ravindran (2008) proved that finding exact symmetry in MDPs is graph-isomorphism complete. However, classical planning algorithms like A* have a fundamental issue with exploiting symmetries. They construct equivalence classes of symmetric states, which explicitly represent states and introduce symmetry breaking. As a result, they are intractable (NP-hard) in maintaining symmetries in trajectory rollout and forward search (for large state spaces and symmetry groups) and are incompatible with differentiable pipelines for representation learning. This limitation hinders their wider applications in reinforcement learning (RL) and robotics.

**State abstraction for detecting symmetries.** Coarsest state abstraction aggregates all symmetric states into equivalence classes, studied in MDP homomorphisms and bisimulation (Ravindran and Barto, 2004; Ferns et al., 2004; Li et al., 2006). However, they require *perfect* MDP dynamics and do not scale up well, typically because of the complexity in maintaining *abstraction mappings* (homomorphisms) and *abstracted MDPs*. van der Pol et al. (2020b) integrate symmetry into model-free RL based on MDP homomorphisms (Ravindran and Barto, 2004) and motivate us to consider planning. Park et al. (2022) learn equivariant transition models, but do not consider planning. Additionally, the typical formulation of symmetric MDPs in (Ravindran and Barto, 2004; van der Pol et al., 2020a; Zhao et al., 2022) is slightly different from our formulation here: we consider sym-

metry between MDPs (rotated maps), instead of within a single MDP. Thus, the reward or transition function additionally depends on map input, as further discussed in Appendix B.2.

**Symmetries and equivariance in deep learning.** Equivariant neural networks are used to incorporate symmetry in supervised learning for different domains (e.g., grids and spheres), symmetry groups (e.g., translations and rotations), and group representations (Bronstein et al., 2021b). Cohen and Welling (2016b) introduce G-CNNs, followed by Steerable CNNs (Cohen and Welling, 2016a) which generalizes from scalar feature fields to vector fields with induced representations. Kondor and Trivedi (2018); Cohen et al. (2020) study the theory of equivariant maps and convolutions. Weiler and Cesa (2021) propose to solve kernel constraints under arbitrary representations for $E(2)$ and its subgroups by decomposing into irreducible representations, named $E(2)$-CNN.

**Differentiable planning.** Our pipeline is based on learning to plan in a neural network in a differentiable manner. Value iteration networks (VIN) (Tamar et al., 2016b) is a representative approach that performs value iteration using convolution on lattice grids, and has been further extended (Niu et al., 2017; Lee et al., 2018; Chaplot et al., 2021; Deac et al., 2021; Zhao et al., 2023). Other than using convolutional networks, works on integrating learning and planning into differentiable networks include Oh et al. (2017); Karkus et al. (2017); Weber et al. (2018); Srinivas et al. (2018); Schrittwieser et al. (2019); Amos and Yarats (2019); Wang and Ba (2019); Guez et al. (2019); Hafner et al. (2020); Pong et al. (2018); Clavera et al. (2020). On the theoretical side, Grimm et al. (2020; 2021) propose to understand the differentiable planning algorithms from a value equivalence perspective.

## 3 BACKGROUND

**Markov decision processes.** We model the path-planning problem as a Markov decision process (MDP) (Sutton and Barto, 2018). An MDP is a 5-tuple $\mathcal{M} = \langle \mathcal{S}, \mathcal{A}, P, R, \gamma \rangle$, with state space $\mathcal{S}$, action space $\mathcal{A}$, transition probability function $P : \mathcal{S} \times \mathcal{A} \times \mathcal{S} \to \mathbb{R}_+$, reward function $R : \mathcal{S} \times \mathcal{A} \to \mathbb{R}$, and discount factor $\gamma \in [0, 1]$. Value functions $V : \mathcal{S} \to \mathbb{R}$ and $Q : \mathcal{S} \times \mathcal{A} \to \mathbb{R}$ represent expected future returns. The core component behind dynamic programming (DP)-based algorithms in reinforcement learning is the *Bellman (optimality) equation* (Sutton and Barto, 2018): $V^*(s) = \max_a R(s, a) + \gamma \sum_{s'} P(s'|s, a) V^*(s')$. Value iteration is an instance of DP to solve MDPs, which iteratively applies the Bellman (optimality) operator until convergence.

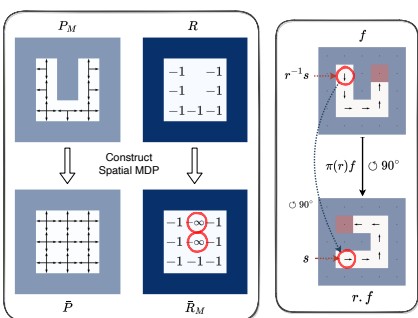

Figure 2: **(Left)** Construction of spatial MDPs from path-planning problems, enabling the use of $G$-invariant transition models. **(Right)** A demonstration of how an action (arrow in red circle) is rotated when a map is rotated.

**Path planning.** The objective of the path-planning problem is to find optimal actions from every location to navigate to the target in shortest time. However, the original path-planning problem is *not equivariant* under *translation* due to obstacles. VINs (Tamar et al., 2016a) implicitly construct an equivalent problem with an *equivariant transition function*, thus CNNs can be used to inject translation equivariance. We visualize the construction of an equivalent "spatial MDP" in Figure 2 (left), where the key idea is to encode obstacle information in the *transition function* from the map (top left) into the *reward function* in the constructed spatial MDP (bottom right) as "trap" states with $-\infty$ reward. Further details about this construction are in Appendices E.1 and E.3. In Figure 2 (right), we provide a visualization of the *representation* $\pi(r)$ of a rotation $r$ of $\circlearrowleft 90°$, and how an action (arrow) is rotated $\circlearrowleft 90°$ accordingly.

**Value Iteration Network.** Tamar et al. (2016a) proposed Value Iteration Networks (VINs) that use a convolutional network to parameterize value iteration. It jointly learns in a latent MDP on a 2D grid, which has the latent reward function $\bar{R} : \mathbb{Z}^2 \to \mathbb{R}^{|\mathcal{A}|}$ and value function $\bar{V} : \mathbb{Z}^2 \to \mathbb{R}$, and applies value iteration on that MDP:

$$\bar{Q}^{(k)}_{\bar{a},i',j'} = \bar{R}_{\bar{a},i',j'} + \sum_{i,j} W^V_{\bar{a},i,j} \bar{V}^{(k-1)}_{i'-i,j'-j} \qquad \bar{V}^{(k)}_{i,j} = \max_{\bar{a}} \bar{Q}^{(k)}_{\bar{a},i,j} \qquad (1)$$

The first equation can be written as: $\bar{Q}^{(k)} = \bar{R}^a + \texttt{Conv2D}(\bar{V}^{(k-1)}; W^V_{\bar{a}})$, where the 2D convolution layer $\texttt{Conv2D}$ has parameters $W^V$.

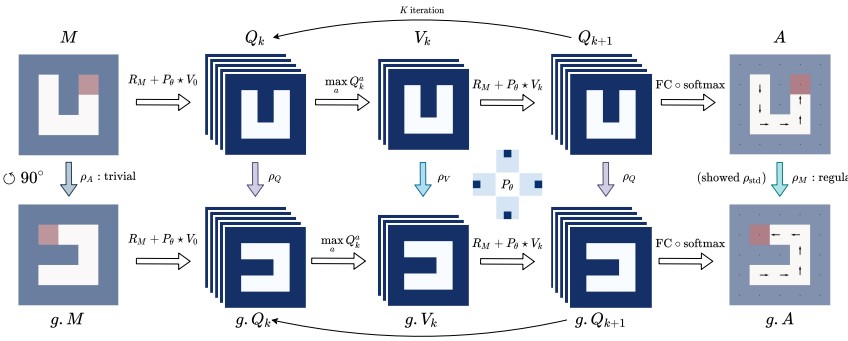

Figure 3: The commutative diagram of Symmetric Value Iteration Network (SymVIN). Every *row* is a full computation graph of VIN. Every *column* rotates the field by ↺ 90°.

We intentionally omit the details on equivariant networks and instead focus on the core idea of integrating symmetry with equivariant networks. We present the necessary group theory background in Appendix C and our full framework and theory in Appendices D and E.

## 4 METHOD: INTEGRATING SYMMETRY INTO PLANNING BY CONVOLUTION

This section presents an algorithmic framework that can provably leverage the inherent symmetry of the path-planning problem in a *differentiable* manner. To make our approach more accessible, we first introduce Value Iteration Networks (VINs) (Tamar et al., 2016a) as the foundation for our algorithm: Symmetric VIN. In the next section, we provide an explanation for why we make this choice and introduce further theoretical guarantees on how to exploit symmetry.

**How to inject symmetry?** VIN uses regular 2D convolutions (Eq. 1), which has *translation equivariance* (Cohen and Welling, 2016b; Kondor and Trivedi, 2018). More concretely, a VIN will output the same value function for the same map patches, up to 2D translation. Characterization of translation equivariance requires a different mechanism and does not *decrease* the search space nor *reduce* a path-planning MDP to an easier problem. We provide a complete description in Appendix E.

Beyond translation, we are more interested in *rotation* and *reflection* symmetries. Intuitively, as shown in Figure 1, if we find the optimal solution to a map, it automatically **generalizes** the solution to all 8 transformed maps (4 rotations times 2 reflections, including identity transformation). This can be characterized by *equivariance* of a planning algorithm Plan, such as value iteration VI: $g.\texttt{Plan}(M) = \texttt{Plan}(g.M)$, where $M$ is a maze map, and $g$ is the symmetry group $D_4$ under which 2D grids are invariant.

More importantly, symmetry also helps **training** of differentiable planning. Intuitively, symmetry in path planning poses additional constraints on its search space: if the goal is in the north, go up; if in the east, go right. In other words, the knowledge can be shared between symmetric cases; the path-planning problem is effectively reduced by symmetry to a smaller problem. This property can also be depicted by equivariance of Bellman operators $\mathcal{T}$, or one step of value iteration: $g.\mathcal{T}[V_0] = \mathcal{T}[g.V_0]$. If we use $\texttt{VI}(M)$ to denote applying Bellman operators on arbitrary initialization until convergence $\mathcal{T}^\infty[V_0]$, value iteration is also equivariant:

$$g.\texttt{VI}(M) \equiv g.\mathcal{T}^\infty[V_0] = \mathcal{T}^\infty[g.V_0] \equiv \texttt{VI}(g.M) \tag{2}$$

We formally prove this equivariance in Theorem 5.1 in next section. In Theorem 5.2, we theoretically show that value iteration in path planning is a specific type of convolution: *steerable convolution* (Cohen and Welling, 2016a). Before that, we first use this finding to present our pipeline on how to use *Steerable CNNs* (Cohen and Welling, 2016a) to integrate symmetry into path planning.

**Pipeline: SymVIN.** We have shown that VI is equivariant given symmetry in path planning. We introduce our method *Symmetric Value Iteration Network* (SymVIN), that realizes equivariant VI by integrating equivariance into VIN with respect to *rotation* and *reflection*, in addition to *translation*. We use an instance of Steerable CNN: $E(2)$-Steerable CNNs (Weiler and Cesa, 2021) and their package e2cnn for implementation, which is equivariant under $D_4$ rotation and reflection, and also $\mathbb{Z}^2$ translation on the 2D grid $\mathbb{Z}^2$. In practice, to inject symmetry into VIN, we mainly need to

replace the translation-equivariant `Conv2D` in Eq. 1 with `SteerableConv`:

$$\bar{Q}_{\bar{a}}^{(k)} = \bar{R}_{\bar{a}} + \texttt{SteerableConv}(\bar{V}; W^V) \qquad\qquad \bar{V}^{(k)} = \max_{\bar{a}} \bar{Q}_{\bar{a}}^{(k)} \qquad (3)$$

We visualize the full pipeline in Figure 3. The map and goal are represented as signals $M : \mathbb{Z}^2 \to \{0,1\}^2$. It will be processed by another layer and output to the core value iteration loop. After some iterations, the final output will be used to predict the actions and compute cross-entropy loss.

Figure 3 highlights our injected equivariance property: if we *rotate* the map (from $M$ to $g.M$), in order to guarantee that the final policy function will also be *equivalently rotated* (from $A$ to $g.A$), we shall guarantee that every *transformation* (e.g., $Q_k \mapsto V_k$ and $V_k \mapsto Q_{k+1}$) in value iteration will also be *equivariant*, for every *pair of columns*. We formally justify our design in the section below and provide more technical details in Appendix E.

**Extension: Symmetric GPPN.** Based on same spirit, we also implement a symmetric version of Gated Path-Planning Networks (GPPN (Lee et al., 2018)). GPPNs use LSTMs to alleviate the issue of unstable gradient in VINs. Although it does not strictly follow value iteration, it still follows the spirit of steerable planning. Thus, we first obtained a fully convolutional variant of GPPN from Kong (2022), called ConvGPPN, which incorporates translational equivariance. It replaces the MLPs in the original LSTM cell with convolutional layers, and then replaces convolutions with equivariant steerable convolutions, resulting in SymGPPN that is equivariant to translations, rotations, and reflections. See Appendix G.1 for details.

## 5 THEORY: VALUE ITERATION IS STEERABLE CONVOLUTION

In the last section, we showed how to exploit symmetry in path planning by equivariance from convolution via intuition. The goal of this section is to (1) connect the theoretical justification with the algorithmic design, and (2) provide intuition for the justification. Even through we focus on a specific task, we hope that the underlying guidelines on integrating symmetry into planning are useful for broader planning algorithms and tasks as well. The complete version is in Appendix E.

**Overview.** There are numerous types of symmetry in various planning tasks. We study symmetry in **path planning** as an example, because it is a straightforward planning problem, and its solutions have been intensively studied in robotics and artificial intelligence (LaValle, 2006; Sutton and Barto, 2018). However, even for this problem, symmetry has *not* been *effectively* exploited in existing planning algorithms, such as Dijkstra's algorithm, A*, or RRT, because it is NP-hard to find symmetric states (Narayanamurthy and Ravindran, 2008).

**Why VIN-based planners?** There are two reasons for choosing value-based planning methods.

1. The expected-value operator in value iteration $\sum_{s'} P(s'|s,a)V(s')$ is (1) *linear* in the value function and (2) *equivariant* (shown in Theorem 5.1). Cohen et al. (2020) showed that any *linear equivariant operator* (on homogeneous spaces, e.g., 2D grid) is a group-convolution operator.

2. Value iteration, using the Bellman (optimality) operator, consists of only maps between signals (steerable fields) over $\mathbb{Z}^2$ (e.g., value map and transition function map). This allows us to inject symmetry by enforcing equivariance to those maps. Taking Figure 1 as an example, the 4 corner states are symmetric under transformations in $D_4$. Equivariance enforces those 4 states to have the same value if we rotate or flip the map. This avoids the need to find if a new state is symmetric to any existing state, which is shown to be NP-hard (Narayanamurthy and Ravindran, 2008).

Our framework for integrating symmetry applies to any value-based planner with the above properties. We found that VIN is conceptually the simplest differentiable planning algorithm that meets these criteria, hence our decision to focus primarily on VIN and its variants.

**Symmetry *from* tasks.** If we want to exploit inherent symmetry in a task to improve planning, there are two major steps: (1) characterize the symmetry in the task, and (2) incorporate the corresponding symmetry into the planning algorithm. The theoretical results in Appendix E.2 mainly characterize the symmetry and direct us to a feasible planning algorithm.

The *symmetry in tasks* for MDPs can be specified by the equivariance property of the transition and reward function, studied in Ravindran and Barto (2004); van der Pol et al. (2020b):

$$\bar{P}(s' \mid s, a) = \bar{P}(g.s' \mid g.s, g.a), \quad \forall g \in G, \forall s, a, s' \tag{4}$$

$$\bar{R}_M(s, a) = \bar{R}_{g.M}(g.s, g.a), \quad \forall g \in G, \forall s, a \tag{5}$$

Note that how the group $G$ *acts* on states and actions is called *group representation*, and is decided by the space $\mathcal{S}$ or $\mathcal{A}$, which is discussed in Equation 19 in Appendix E.2. We emphasize that the equivariance property of the reward function is different from prior work (Ravindran and Barto, 2004; van der Pol et al., 2020b): in our case, the reward function encodes obstacles as well, and thus depends on map input $M$. Intuitively, using Figure 1 as an example, if a position $s$ is rotated to $g.s$, in order to find the correct original reward $R$ before rotation, the input map $M$ must also be rotated $g.M$. See Appendix E for more details.

**Symmetry *into* planning.** As for exploiting the *symmetry in planning algorithms*, we focus on value iteration and the VIN algorithm. We first prove in Theorem 5.1 that value iteration for path planning respects the *equivariance* property, motivating us to incorporate symmetry with equivariance.

**Theorem 5.1** (informal). *If transition is $G$-invariant, expected-value operator $\sum_{s'} P(s'|s,a)V(s')$ and value iteration are equivariant under translation, rotation, reflection on the 2D grid.*

We visualize the equivariance of the central value-update step $R + \gamma P \star V_k$ in Figure 4. The upper row is a value field $V_k$ and its rotated version $g.V_k$, and the lower row is for $Q$-value fields $Q_k$ and $g.Q_k$. The diagram shows that, if we input a rotated value $g.V_k$, the output $R + \gamma P \star g.V_k$ is guaranteed to be equal to rotated $Q$-field $g.Q_k$. Additionally, rotating $Q$-field $g.Q_k$ has two components: (1) spatially rotating each grid (a feature channel for an action $Q(\cdot, a)$) and (2) cyclically permuting the channels (black arrows). The red dashed line points out how a specific grid of a $Q$-value grid $Q_k(\cdot, \text{South})$ got rotated and permuted. We discuss the theoretical guarantees in Theorem 5.1 and provide full proofs in Appendix F.

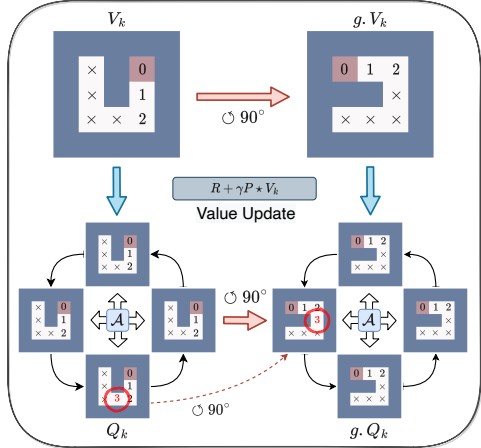

Figure 4: Commutative diagram of a single step of value update, showing equivariance under rotations. Each grid in the $Q$-value field corresponds to all the values $Q(\cdot, a)$ of a single action $a$.

However, while the first theorem provides intuition, it is inadequate since it only shows the equivariance property for *scalar-valued* transition probabilities and value functions, and does not address the implementation of VINs with *multiple feature channels* in CNNs. To address this gap, the next theorem further proves that value iteration is a general form of *steerable convolution*, motivating the use of steerable CNNs by Cohen and Welling (2016a) to replace regular CNNs in VIN. This is related to Cohen et al. (2020) that proves steerable convolution is the most general linear equivariant map on homogeneous spaces.

**Theorem 5.2** (informal). *If transition is $G$-invariant, the expected-value operator is expressible as a steerable convolution $\star$, which is equivariant under translation, rotation, and reflection on 2D grid. Thus, value iteration (with $\max$, $+$, $\times$) is a form of steerable CNN (Cohen and Welling, 2016a).*

We provide a complete version of the framework in Section E and the proofs in Section F. This justifies why we should use Steerable CNN (Cohen and Welling, 2016a): the VI itself is composed of steerable convolution and additional operations ($\max$, $+$, $\times$). With equivariance, the value function $\mathbb{Z}^2 \to \mathbb{R}$ is $D_4$-invariant, thus the planning is effectively done in the quotient space reduced by $D_4$.

**Summary.** We study how to inject symmetry into VIN for (2D) path planning, and expect the task-specific technical details are useful for two types of readers. *(i) Using VIN.* If one uses VIN for differentiable planning, the resulting algorithms SymVIN or SymGPPN can be a plug-in alternative, as part of a larger end-to-end system. Our framework generalizes the idea behind VINs and enables us to understand its applicability and restrictions. *(ii) Studying path planning.* The proposed framework characterizes the symmetry in path planning, so it is possible to apply the underlying ideas to other domains. For example, it is possible to extend to higher-dimensional continuous Euclidean

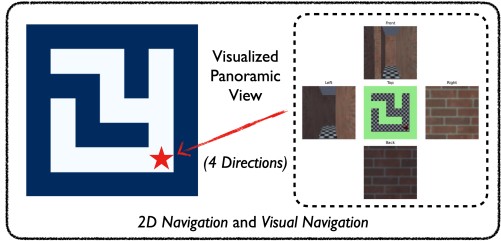 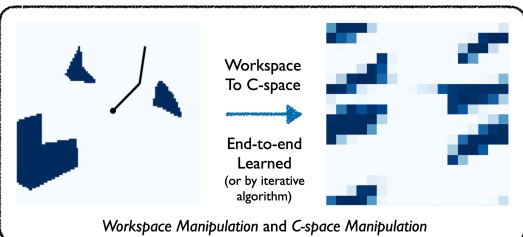

Figure 5: **(Left)** We randomly generate occupancy grid $\mathbb{Z}^2 \to \{0, 1\}$ for **2D navigation**. For **visual navigation**, each position provides $32 \times 32 \times 3$ egocentric panoramic RGB images in 4 directions for each location $\mathbb{Z}^2 \to \mathbb{R}^{4 \times 32 \times 32 \times 3}$. One location is visualized. **(Right)** The top-down view (left) is the *workspace* of a 2-DOF manipulation task. For **workspace manipulation**, it is converted by a mapper layer to configuration space, shown in the right subfigure. For **C-space manipulation**, the ground-truth C-space is provided to the planner.

spaces or spatial graphs (Weiler et al., 2018; Brandstetter et al., 2021). Additionally, we emphasize that the *symmetry in spatial MDPs* is different from *symmetric MDPs* (Zinkevich and Balch, 2001; Ravindran and Barto, 2004; van der Pol et al., 2020a), since our reward function is *not* $G$-invariant (if not conditioning on obstacles). We further discuss this in Appendices B.2 and E.4.

## 6 EXPERIMENTS

We experiment with VIN, GPPN and our SymPlan methods on four path-planning tasks, using both *given* and *learned* maps. Additional experiments and ablation studies are in Appendix H.

**Environments and datasets.** We demonstrate the idea in four path-planning tasks: (1) **2D navigation**, (2) **visual navigation**, (3) 2 degrees of freedom (2-DOF) **configuration-space (C-space) manipulation**, and (4) **2-DOF workspace manipulation**. For each task, we consider using either *given* (2D navigation and 2-DOF configuration-space manipulation) or *learned* maps (visual navigation and 2-DOF workspace manipulation). In the latter case, the planner needs to jointly learn a mapper that converts egocentric panoramic images (visual navigation) or workspace states (workspace manipulation) into a map that the planners can operate on, as in (Lee et al., 2018; Chaplot et al., 2021). In both cases, we randomly generate training, validation and test data of $10K/2K/2K$ maps for all map sizes, to demonstrate data efficiency and generalization ability of symmetric planning. Due to the small dataset sizes, test maps are unlikely to be symmetric to the training maps by any transformation from the symmetry groups $G$. For all environments, the planning domain is the 2D regular grid as in VIN, GPPN and SPT $\mathcal{S} = \Omega = \mathbb{Z}^2$, and the action space is to move in 4 $\circlearrowleft$ directions[1]: $\mathcal{A} = (\text{north}, \text{ west}, \text{ south}, \text{ east})$.

**Methods: planner networks.** We compare five planning methods, two of which are our SymPlan methods. Our two equivariant methods are based on *Value Iteration Networks* (**VIN**, (Tamar et al., 2016a)) and *Gated Path Planning Networks* (**GPPN**, (Lee et al., 2018)). Our equivariant version of VIN is named **SymVIN**. For GPPN, we first obtained a *fully convolutional* version, named **ConvGPPN** (Kong, 2022), and furthermore obtain **SymGPPN** by replacing standard convolutions with steerable CNNs. All methods use (equivariant) convolutions with *circular padding* to plan in configuration spaces for the manipulation tasks, except GPPN which is not fully convolutional.

**Training and evaluation.** We report success rate and training curves over 3 seeds. The training process (on given maps) follows (Tamar et al., 2016a; Lee et al., 2018), where we train 30 epochs with batch size 32, and use kernel size $F = 3$ by default. The default batch size is 32. GPPN variants need smaller number because LSTM consumes much more memory.

### 6.1 PLANNING ON GIVEN MAPS

**Environmental setup.** In the **2D navigation** task, the map and goal are randomly generated, where the map size is $\{15, 28, 50\}$. In **2-DOF manipulation** in configuration space, we adopt the setting

---

[1]Note that the MDP action space $\mathcal{A}$ needs to be *compatible* with the group action $G \times \mathcal{A} \to \mathcal{A}$. Since the E2CNN package (Weiler and Cesa, 2021) uses *counterclockwise* rotations $\circlearrowleft$ as generators for rotation groups $C_n$, the action space needs to be *counterclockwise* $\circlearrowleft$.

Table 1: Averaged test success rate (%) for using 10K/2K/2K dataset for all four types of tasks.

| Method (10K Data) | Navigation | | | Visual | Manipulation | | Workspace |
|---|---|---|---|---|---|---|---|
| | $15 \times 15$ | $28 \times 28$ | $50 \times 50$ | | $18 \times 18$ | $36 \times 36$ | |
| VIN | 66.97 | 67.57 | 57.92 | 50.83 | 77.82 | 84.32 | 80.44 |
| **SymVIN** | 98.99 | 98.14 | 86.20 | 95.50 | 99.98 | 99.36 | **91.10** |
| GPPN | 96.36 | 95.77 | 91.84 | 93.13 | 2.62 | 1.68 | 3.67 |
| ConvGPPN | 99.75 | 99.09 | 97.21 | 98.55 | 99.98 | 99.95 | 89.88 |
| **SymGPPN** | **99.98** | **99.86** | **99.49** | **99.78** | **100.00** | **99.99** | 90.50 |

in Chaplot et al. (2021) and train networks to take as input "maps" in configuration space, represented by the state of the two manipulator joints. We randomly generate $0$ to $5$ obstacles in the manipulator workspace. Then the 2 degree-of-freedom (DOF) configuration space is constructed from the workspace and discretized into 2D grids with sizes $\{18, 36\}$, corresponding to bins of $20°$ and $10°$, respectively. All methods are trained using the same network size, where for the equivariant versions, we use *regular* representations for all layers, which has size $|D_4| = 8$. We keep the same parameters for all methods, so all equivariant convolution layers with *regular* representations will have higher embedding sizes. Due to memory constraints, we use $K = 30$ iterations for 2D maze navigation, and $K = 27$ for manipulation. We use kernel sizes $F = \{3, 5, 5\}$ for $m = \{15, 28, 50\}$ navigation, and $F = \{3, 5\}$ for $m = \{18, 36\}$ manipulation.

**Results.** We show the averaged test results for both 2D navigation and C-space manipulation tasks on generalizing to unseen maps (Table 1) and the training curves for 2D navigation (Figure 6). For VIN-based methods, SymVIN learns much faster than standard VIN and achieves almost perfect asymptotic performance. For GPPN-based methods, we found the translation-equivariant convolutional variant ConvGPPN works better than the original one in (Lee et al., 2018), especially in learning speed. SymVIN fluctuates in some runs. We observe that the first epoch has abnormally high loss in most runs and quickly goes down in the second or third epoch, which indicates effects from initialization.

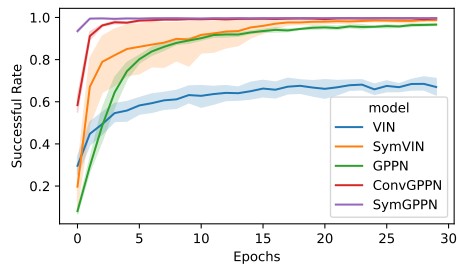

Figure 6: Training curves on 2D navigation with 10K of $15 \times 15$ maps. Faded areas indicate standard error.

SymGPPN further boosts ConvGPPN and outperforms all other methods and does not experience variance. One exception is that standard GPPN learns poorly in C-space manipulation. For GPPN, the added circular padding in the convolution encoder leads to a gradient-vanishing problem.

Additionally, we found that using regular representations (for $D_4$ or $C_4$) for state value $V : \mathbb{Z}^2 \to \mathbb{R}^{C_V}$ (and for $Q$-values) works better than using trivial representations. This is counterintuitive since we expect the $V$ value to be scalar $\mathbb{Z}^2 \to \mathbb{R}$. One reason is that switching between regular (for $Q$) and trivial (for $V$) representations introduces an unnecessary bottleneck. Depending on the choice of representations, we implement different max-pooling, with details in Appendix G.2. We also empirically found using FC only in the final layer $Q_K \mapsto A$ helps stabilize the training. The ablation study on this and more are described in Appendix H.

**Remark.** Our two symmetric planners are both significantly better than their standard counterparts. Notably, we did not include any symmetric maps in the test data, which symmetric planners would perform much better on. There are several potential sources of advantages: (1) SymPlan allows parameter sharing across positions and maps and implicitly enables planning in a reduced space: every $(s, a, s')$ generalizes to $(g.s, g.a, g.s')$ for any $g \in G$, (2) thus it uses training data more efficiently, (3) it reduces the hypothesis-class size and facilitates generalization to unseen maps.

## 6.2 PLANNING ON LEARNED MAPS: SIMULTANEOUSLY PLANNING AND MAPPING

**Environmental setup.** For **visual navigation**, we randomly generate maps using the same strategy as before, and then render four egocentric panoramic views for each location from 3D environments produced with *Gym-MiniWorld* (Chevalier-Boisvert, 2018), which can generate 3D mazes with any

layout. For $m \times m$ maps, all egocentric views for a map are represented by $m \times m \times 4$ RGB images. For **workspace manipulation**, we randomly generate 0 to 5 obstacles in workspace as before. We use a mapper network to convert the $96 \times 96$ workspace (image of obstacles) to the $m \times m$ 2 degree-of-freedom (DOF) configuration space (2D occupancy grid). In both environments, the setup is similar to Section 6.1, except here we only use map size $m = 15$ for visual navigation (training for 100 epochs), and map size $m = 18$ for workspace manipulation (training for 30 epochs).

**Methods: mapper networks and setup.** For **visual navigation**, we implemented an equivariant mapper network based on Lee et al. (2018). The mapper network converts every image into a 256-dimensional embedding $m \times m \times 4 \times 256$ and then predicts the map layout $m \times m \times 1$. For **workspace manipulation**, we use a U-net (Ronneberger et al., 2015) with residual-connection (He et al., 2015) as a mapper. For more training details, see Appendix H.

**Results.** The results are shown in Table 1, under the columns "Visual" (navigation, $15 \times 15$) and "Workspace" (manipulation, $18 \times 18$). In visual navigation, the trends are similar to the 2D case: our two symmetric planners both train much faster. Besides standard VIN, all approaches eventually converge to near-optimal success rates during training (around 95%), but the validation and test results show large performance gaps (see Figure 23 in Appendix H). SymGPPN has almost no generalization gap, whereas VIN does not generalize well to new 3D visual navigation environments. SymVIN significantly improves test success rate and is comparable with GPPN. Since the input is raw images and a mapper is learned end-to-end, it is potentially a source of generalization gap for some approaches. In workspace manipulation, the results are similar to our findings in C-space manipulation, but our advantage over baselines were smaller. We found that the mapper network was a bottleneck, since the mapping for obstacles from workspace to C-space is non-trivial to learn.

### 6.3 Results on generalization to larger maps

To demonstrate the generalization advantage of our methods, all methods are trained on small maps and tested on larger maps. All methods are trained on $15 \times 15$ with $K = 30$. Then we test all methods on map size $15 \times 15$ through $99 \times 99$, averaging over 3 seeds (3 model checkpoints) for each method and 1000 maps for each map size. Iterations $K$ were set to $\sqrt{2}M$, where $M$ is the test map size (x-axis). The results are shown in Figure 7.

**Results.** SymVIN generalizes better than VIN, although the variance is greater. GPPN tends to diverge for larger values of $K$. ConvGPPN converges, but it fluctuates for different seeds. SymGPPN shows the best generalization and has small variance. In conclusion, SymVIN and SymGPPN generalize better to different map sizes, compared to all non-equivariant baselines.

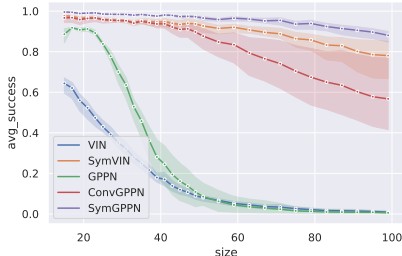

**Remark.** In summary, our results show that the Sym-Plan models demonstrate end-to-end planning and learning ability, potentially enabling further applications to other tasks as a differentiable component for planning. Additional results and ablation studies are in Appendix H.

Figure 7: Results for testing on larger maps, when trained on size 15 map. Our methods outperform all baselines.

## 7 Discussion

In this work, we study the symmetry in the 2D path-planning problem, and build a framework using the theory of steerable CNNs to prove that value iteration in path planning is actually a form of steerable CNN (on 2D grids). Motivated by our theory, we proposed two symmetric planning algorithms that provided significant empirical improvements in several path-planning domains. Although our focus in this paper has been on $\mathbb{Z}^2$, our framework can potentially generalize to path planning on higher-dimensional or even continuous Euclidean spaces (Weiler et al., 2018; Brandstetter et al., 2021), by using *equivariant operations* on *steerable feature fields* (such as steerable convolutions, pooling, and point-wise non-linearities) from steerable CNNs. We hope that our SymPlan framework, along with the design of practical symmetric planning algorithms, can provide a new pathway for integrating symmetry into differentiable planning.

## 8 ACKNOWLEDGEMENT

This work was supported by NSF Grants #2107256 and #2134178. R. Walters is supported by The Roux Institute and the Harold Alfond Foundation. We also thank the audience from previous poster and talk presentations for helpful discussions and anonymous reviewers for useful feedback.

## 9 REPRODUCIBILITY STATEMENT

We provide additional details in the appendix. We also plan to open source the codebase. We briefly outline the appendix below.

1. Additional Discussion
2. Background: Technical background and concepts on steerable CNNs and group CNNs
3. Method: we provide full details on how to reproduce it
4. Theory/Framework: we provide the complete version of the theory statements
5. Proofs: this includes all proofs
6. Experiment / Environment / Implementation details: useful details for reproducibility
7. Additional results

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

CONTENTS

# A   OUTLINE

We provide a table of content above.

We omit technical details on symmetry and equivariant networks in the main paper and delay them here. Specifically, for readers interested in additional details on how to use equivariant networks for symmetric planning, we recommend an order as follows: **(1) Basics on group representations and equivariant networks in Section C.1. (2) Practice on building SymVIN in Section D.1. (3) Detailed formulation on SymPlan in Section E.1 and Section E.2.**

The rest technical sections provide additional reading materials for the readers interested in more in-depth account on studying symmetry in reinforcement learning and planning.

# B   ADDITIONAL DISCUSSION

## B.1   LIMITATIONS AND EXTENSIONS

**Assumption on known domain structure.**   As in VIN, although the framework of steerable planning can potentially handle different domains, one important hidden assumption is that the underlying domain $\Omega$ (state space), is known. In other words, we fix the structure of learned transition kernels $p(s' \mid s, a)$ and estimate coefficients of it. One potential method is to use Transformers that learn attention weights to all states in $\mathcal{S}$, which has been partially explored in SPT (Chaplot et al., 2021). Additionally, it is also possible to treat unknown MDPs as learned transition graphs, as explored in XLVIN (Deac et al., 2021). We leave the consideration of symmetry in unknown underlying domains for future work.

**The curse of dimensionality.**   The paradigm of steerable planning still requires full expansion in computing value iteration (opposite to *sampling-based*), since we realize the symmetric planner using group equivariant convolutions (essentially summation or integral). Convolutions on high-dimensional space could suffer from the curse of dimensionality for higher dimensional domains, and are vastly under-explored. This is a primary reason why we need sampling-based planning algorithms. If the domain (state-action transition graph) is sparsely connected, value iteration can still scale up to higher dimensions. It is also unclear either when steerable planning would fail, or how sampling-based algorithms could be integrated with the symmetric planning paradigm.

## B.2   THE CONSIDERED SYMMETRY AND DIFFERENCE TO EXISTING WORK

We need to differentiate between two types of symmetry in MDPs. Let's take spatial graph as illustrative example to understand the potential symmetry from a higher level, which means that the nodes $\mathcal{V}$ in the graph have spatial coordinates $\mathbb{Z}^n$ or $\mathbb{R}^n$. Our 2D path planning is a special case of spatial graph, where the actions can only move to adjacent spatial nodes.

Let the graph denoted as $\mathcal{G} = \langle \mathcal{V}, \mathcal{E} \rangle$. $\mathcal{E}$ is the set of edges connecting two states with an action. One type of symmetry is the symmetry of the graph itself. For the grid case, it means that after $D_4$ rotation or reflection, the map is unchanged.

Another type of symmetry comes from the isometries of the space. For a spatial graph, we can rotate it freely in a space, while the relative positions are unchanged. For our grid case, it is shown in the Figure 1 that rotating a map resulting in the rotated policy. However, the map or policy itself can never be equal under any transformation in $D_4$.

In other words, the first type is symmetry within a MDP (rely on the property of the MDP itself $\mathcal{M}$, or $\mathrm{Aut}(\mathcal{M})$), and the second type is symmetry between MDPs (only rely on the property of the underlying spatial space $\mathbb{Z}^2$, or $\mathrm{Aut}(\mathbb{Z}^2)$).

Nevertheless, we could input map $M$ and somehow treat symmetric states between MDPs as one state. See the proofs section for more details.

### B.3 Additional Discussions

**Potential generalization to 3D space.** The goal of our work is to integrate symmetry into differentiable planning. While there is a rich branch in math and physics that studies representation theory, we use 2D settings to keep the representation part minimal and provide an example pipeline for using equivariant networks to integrate symmetry. For 3D and other cases, the key is still equivariance, but they require more technical attention on the equivariant network side.

Representation theory has extensively studied 3D rotations. SO(3)-equivariant networks have been developed in the equivariant network field by researchers like Cohen et al. (2018) (Spherical CNNs) and Geiger and Smidt (2022) (2022, e3nn: Euclidean Neural Networks). SO(3)-equivariant networks avoid gimbal lock by not predicting pose but only needing equivariance to SO(3). SO(3) elements only appear in solving equivariant kernel constraints.

**Inference on realistic robotic maps.** Our choice of small and toyish maps ($100 \times 100$ or smaller) is in line with prior work, such as VIN, GPPN, and SPT, which mainly experimented on $15 \times 15$ and $28 \times 28$ maps. While we recognize that in robotics planning, maps can be significantly larger, we believe that integrating symmetry into differentiable planning is an orthogonal topic with the scalability of differentiable planning algorithms. However, our visual navigation experiment shows that our method can learn from panoramic egocentric RGB images of all locations, and our generalization experiment demonstrates the potential of scalability as the model can generalize to larger maps, which has not been explored in prior work. In comparison with robotic navigation works like Active Neural SLAM, our approach aims to improve only the differentiable planning module, while it is a systematic pipeline with a hierarchical architecture of global and local planners. Our Symmetric Planners could serve as an alternative to either the global or local planner, but we are not comparing our approach with the whole Active Neural SLAM pipeline. Additionally, Neural SLAM uses room-like maps, which have a different structure and require less planning horizon compared to our maze-like maps.

**Measure of efficiency of symmetry in path planning.** In Appendix Section E.4, we demonstrate how symmetry can aid path planning and provide intuition for our approach. We show that the MDPs from rotated maps are related by MDP homomorphisms, which has been previously used in Ravindran and Barto (2004). This means that states related by $D_4$ rotations/reflections can be aggregated into one state. We inject this symmetry property into the SymVIN algorithm by enforcing $D_4$-invariance of the value function on $\mathbb{Z}^2$, which is equivalent to a function on the quotient space. The global and local equivariance to $D_4$ enables our Symmetric Planning algorithms to effectively plan on the quotient/reduced MDP with a smaller state space. By local equivariance, we mean that every value iteration step is a convolution (or other equivariant) operation and is equivariant to rotation/reflection (see Figure 3 and 4), thus the search space in planning can be exponentially shrunk with respect to the planning horizon. Although we haven't formally proven this or the sample complexity side, intuitively, this gives $|G|$ times smaller state space and sample complexity.

Equivariance's benefits in general supervised learning tasks are still being explored. Recent work includes showing improved generalization bounds for group invariant/equivariant deep networks, as demonstrated in the work of Sannai et al. (2021), Elesedy and Zaidi (2021), and Behboodi et al. (2022), among others.

## C Background: Equivariant Networks

We omit technical details in the main paper and delay them here. This section introduces the background on equivariant networks and representation theory. The first subsection covers necessary basics, while the rest subsections provide additional reading materials for the readers interested in more in-depth account on the preliminaries on studying symmetry in reinforcement learning and planning.

### C.1 Basics: Groups and Group Representations

**Symmetry groups and equivarance.** A symmetry *group* is defined as a set $G$ together with a binary composition map satisfying the axioms of associativity, identity, and inverse. A (left) *group action*

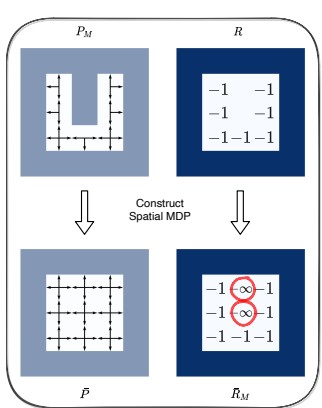 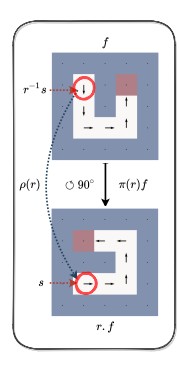 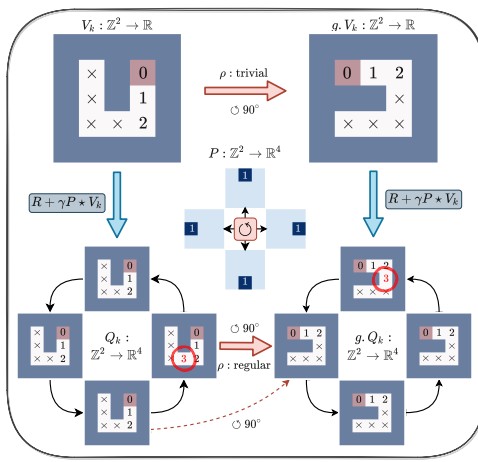

Figure 8: **(Left)** Construction of spatial MDPs from path planning problems, enabling $G$-invariant transition. **(Middle)** The group acts on a feature field (MDP actions). We need to find the element in the original field by $f(r^{-1}x)$, and also rotate the arrow by $\rho(r)$, where $r \in D_4$. We represent one-hot actions as arrows (vector field, using $\rho_{\text{std}}$) for visualization. **(Right)** Equivariance of $V \mapsto Q$ in Bellman operator on feature fields, under $\circlearrowleft 90° \in C_4$ rotation, which visually explains Theorem E.1. The example simulates VI for one step (see red circles; minus signs omitted) with true transition $P$ using $\circlearrowleft$ N-W-S-E actions. The $Q$-value field are for 4 actions and can be viewed as either $\mathbb{Z}^2 \to \mathbb{R}^4$ ((Cohen and Welling, 2016a; Weiler and Cesa, 2021)) or $\mathbb{Z}^2 \rtimes C_4 \to \mathbb{R}$ (on $p4$ group, (Cohen and Welling, 2016b)). Simplified figures are presented in the main paper.

of $G$ on a set $\mathcal{X}$ is defined as the mapping $(g, x) \mapsto g.x$ which is compatible with composition. Given a function $f : \mathcal{X} \to \mathcal{Y}$ and $G$ acting on $\mathcal{X}$ and $\mathcal{Y}$, then $f$ is $G$-*equivariant* if it commutes with group actions: $g.f(x) = f(g.x), \forall g \in G, \forall x \in \mathcal{X}$. In the special case the action on $\mathcal{Y}$ is trivial $g.y = y$, then $f(x) = f(g.x)$ holds, and we say $f$ is $G$-*invariant*.

**Group representations.** We mainly use two groups: dihedral group $D_4$ and cyclic group $C_4$. The cyclic group of 4 elements is $C_4 = \langle r \mid r^4 = 1 \rangle$, a symmetry group of rotating a square. The dihedral group $D_4 = \langle r, s \mid r^4 = s^2 = (sr)^2 = 1 \rangle$ includes both rotations $r$ and reflections $s$, and has size $|D_4| = 8$. A group representation defines how a group action transforms a vector space $G \times S \to S$. These groups have three types of representations of our interest: *trivial, regular*, and *quotient* representations, see (Weiler and Cesa, 2021). The *trivial representation* $\rho_{\text{triv}}$ maps each $g \in G$ to 1 and hence fixes all $s \in S$. The *regular representation* $\rho_{\text{reg}}$ of $C_4$ group sends each $g \in C_4$ to a $4 \times 4$ permutation matrix that cyclically permutes a 4-element vector, such as a one-hot 4-direction action. The regular representation of $D_4$ maps each element to an $8 \times 8$ permutation matrix which does not act on 4-direction actions, which requires the *quotient representations* (quotienting out $sr^2$ reflection part) and forming a $4 \times 4$ permutation matrix. It is worth mentioning the *standard representation* of the cyclic groups, which are $2 \times 2$ rotation matrices, only used for visualization (Figure 8 middle).

**Steerable feature fields and Steerable CNNs.** The concept of *feature fields* is used in (equivariant) CNNs (Bronstein et al., 2021a; Cohen et al., 2020; Kondor and Trivedi, 2018; Cohen and Welling, 2016a;b; Weiler and Cesa, 2021). The pixels of an 2D RGB image $x : \mathbb{Z}^2 \to \mathbb{R}^3$ on a domain $\Omega = \mathbb{Z}^2$ is a feature field. In steerable CNNs for 2D grid, features are formed as *steerable feature fields* $f : \mathbb{Z}^2 \to \mathbb{R}^C$ that associate a $C$-dimensional feature vector $f(x) \in \mathbb{R}^C$ to each element on a base space, such as $\mathbb{Z}^2$. Defined like this, we know how to transform a steerable feature field and also the feature field after applying CNN on it, using some group (Cohen and Welling, 2016a). The type of CNNs that operates on steerable feature fields is called Steerable CNN (Cohen and Welling, 2016a), which is equivariant to groups including *translations* as subgroup $(\mathbb{Z}^2, +)$, extending (Cohen and Welling, 2016b). It needs to satisfy a *kernel steerability* constraint, where the $\mathbb{R}^2$ and $\mathbb{Z}^2$ cases are considered in (Weiler and Cesa, 2021). We consider the 2D grid as our domain $\Omega = \mathcal{S} = \mathbb{Z}^2$ and use $G = p4m$ group as the running example. The group $p4m = (\mathbb{Z}^2, +) \rtimes D_4$ (wallpaper group) is semi-direct product of discrete translation group $\mathbb{Z}^2$ and dihedral group $D_4$, see (Cohen and Welling, 2016b;a). We visualize the *transformation law* of $p4m$ on a feature field on $\Omega = \mathbb{Z}^2$ in

Figure 9: Visualization of the permutation representations of $D_4$ group for every element $g \in D_4$ (4 rotations each row and 2 reflections each column). They are (1) the trivial representation, (2) the regular representation, (3) the quotient representation (quotienting out *rotations*), (4) the quotient representation (quotienting out *reflections*).

**Figure 8 (Middle)**, usually referred as *induced representation* (Cohen and Welling, 2016a; Weiler and Cesa, 2021).

## C.2 GROUP REPRESENTATIONS: VISUAL UNDERSTANDING

A group representation is a (linear) group action that defines how a group acts on some space. Cohen and Welling (2016b;a); Weiler and Cesa (2021) provide more formal introduction to them in the context of equivariant neural networks. We provide visual understanding and refer the readers to them for comprehensive account.

To visually understand how the group $D_4$ acts on some vector space, we visualize the trivial, regular, and quotient (quotienting out reflections $sr^2$) representations, which are *permutation matrices*. If we apply such a representation $\rho(g)(g \in D_4)$ to a vector, the elements get *cyclically permuted*. See Figure 9.

The quotient representation that quotients out reflections and has dimension $4 \times 4$ is what we need to use on the 4-direction action space.

## C.3 GEOMETRIC DEEP LEARNING

We review another set of important concepts that motivate our formulation of steerable planning: geometric deep learning and the theories on connecting equivariance and convolution (Bronstein et al., 2021a; Cohen et al., 2020; Kondor and Trivedi, 2018). Bronstein et al. (2021a) use $x$ for feature fields while Cohen and Welling (2016a); Cohen et al. (2020); Weiler and Cesa (2021) use $f$.

**Convolutional feature fields.** The signals are taken from set $\mathcal{C} = \mathbb{R}^D$ on some structured *domain* $\Omega$, and all mappings from the domain to signals forms the space of $\mathcal{C}$-valued signals $\mathcal{X}(\Omega, \mathcal{C}) = \{f : \Omega \to \mathcal{C}\}$, or $\mathcal{X}(\Omega)$ for abbreviation. For instance, for RGB images, the domain is the 2D $n \times n$ grid $\Omega = \mathbb{Z}_n \times \mathbb{Z}_n$, and every pixel can take RGB values $\mathcal{C} = \mathbb{R}^3$ at each point in the domain $u \in \Omega$, represented by a mapping $x : \mathbb{Z}_n \times \mathbb{Z}_n \to \mathbb{R}^3$. A function on images thus operates on $3n^2$-dimensional inputs.

It is argued that the underlying geometric structure of domains $\Omega$ plays key role in alleviating the curse of dimensionality, such as convolution networks in computer vision, and this framework is named *Geometric Deep Learning*. We refer the readers to Geometric Deep Learning (Bronstein et al., 2021a) for more details, and to more rigorous theories on the relation between equivariant maps and convolutions in (Cohen et al., 2020) (vector fields through induced representations) and (Kondor and Trivedi, 2018) (scalar fields through trivial representations).

**Group convolution.** Convolutions are shift-equivariant operations, and vice versa. This is the special case for $\Omega = \mathbb{R}$, which can be generalized to any group $G$ (that we can integrate or sum over). The *group convolution* for signals on $\Omega$ is then defined[2] as

$$(f \star \psi)(g) = \langle f, \rho(g)\psi \rangle = \int_\Omega f(u)\psi(g^{-1}u)\mathrm{d}u, \tag{6}$$

where $\psi(u)$ is shifted copies of a filter, usually locally supported on a subset of $\Omega$ and padded outside. Note that although $x$ takes $u \in \Omega$, the feature map $(x \star \psi)$ takes as input the elements $g \in G$ instead of points on the domain $u \in \Omega$. All following group convolution layers take $G$:

$\mathcal{X}(G) \to \mathcal{X}(G)$. In the grid case, the domain $\Omega$ is *homogeneous* space of the group $G$, i.e. the group $G$ acts transitively: for any two points $u, v \in \Omega$ there exists a symmetry $g \in G$ to reach $u = gv$.

Analogous to classic shift-equivariant convolutions, the generalized group convolution is $G$-equivariant (Cohen et al., 2020). It is observed that $\langle x, \rho(g)\theta \rangle = \langle \rho(g^{-1})x, \theta \rangle$, and from the defining property of group representations $\rho(h^{-1})\rho(g) = \rho(h^{-1}g)$, the $G$-equivariance of group convolution follows (Bronstein et al., 2021a):

$$(\rho(h)x \star \theta)(g) = \langle \rho(h)x, \rho(g)\theta \rangle = \left\langle x, \rho(h^{-1}g)\theta \right\rangle = \rho(h)(x \star \theta)(g) \tag{7}$$

**Steerable convolution kernels.** Steerable convolutions extend group convolutions to more general setup and decouple the computation cost with the group size (Cohen and Welling, 2016a; Cohen, 2021). For example, $E(2)$-steerable CNNs (Weiler and Cesa, 2021) apply it for $E(2)$ group, which is semi-direct product of translations $\mathbb{R}^2$ and a fiber group $H$, where $H$ is a group of transformations that fixes the origin and is $O(2)$ or its subgroups. The representation on the signals/fields is induced from a representation of the fiber group $H$. Use $\mathbb{R}^2$ as example, a steerable kernel only needs to be $H$-equivariant by satisfying the following constraint (Weiler and Cesa, 2021):

$$\psi(hx) = \rho_{\text{out}}(h)\psi(x)\rho_{\text{in}}(h^{-1}) \quad \forall h \in H, x \in \mathbb{R}^2. \tag{8}$$

## C.4   STEERABLE CNNS

We still use the running example on $\mathbb{Z}^2$ and group $p4m = \mathbb{Z}^2 \rtimes D_4$.

**Induced representations.** We follow (Cohen and Welling, 2016a; Cohen et al., 2020) to use $\pi$ for *induced* representations. We still use feature fields over $\mathbb{Z}^2$ as example.

As shown in **Figure 8 middle**, to transform a feature field $f : \mathbb{Z}^2 \to \mathbb{R}^C$ on base $\mathbb{Z}^2$ with group $p4m = \mathbb{Z}^2 \rtimes D_4$, we need the *induced representation* (Cohen and Welling, 2016a; Cohen et al., 2020). The induced representation in this case is denoted as $\pi(g) \triangleq \text{ind}_{D_4}^{\mathbb{Z}^2 \rtimes D_4} \rho(g)$ (for all $g$), which means how the group action of $D_4$ transforms a feature field on $\mathbb{Z}^2 \rtimes D_4$.

It acts on the feature field with two parts: (1) on the base space $\mathbb{Z}^2$ and (2) on the fibers (feature channels $\mathbb{R}^C$) by fiber group $H = D_4$ (Cohen and Welling, 2016a; Weiler and Cesa, 2021). More specifically, applying a translation $t \in \mathbb{Z}^2$ and a transformation $r \in D_4$ to some field $f$, we get $\pi(tr)f$ (Cohen and Welling, 2016a; Weiler and Cesa, 2021):

$$f(x) \mapsto [\pi(tr)f](x) \triangleq \rho(r) \cdot \left[ f\left((tr)^{-1}x\right) \right]. \tag{9}$$

$\rho(r)$ is the fiber representation that transforms the fibers $\mathbb{R}^C$, and $(tr)^{-1}x$ finds the element before group action (or equivalently transforming the base space $\mathbb{Z}^2$). Thus, $\pi$ only depends on the fiber representation $\rho$ but not the latter part, thus named *induced representation* by $\rho$.

**Steerable convolution vs. group convolution.** The steerable convolution on $\mathbb{Z}^2$ The understanding of this point helps to understand how a group acts on various feature fields and the design of state space for path planning problems. We use the discrete group $p4 = \mathbb{Z}^2 \rtimes C_4$ as example, which consists of $\mathbb{Z}^2$ translations and $90°$ rotations. The only difference with $p4m$ is $p4$ does not have reflections.

The group convolution with filter $\psi$ and signal $x$ on grid (or $\mathbf{p} \in \mathbb{Z}^2$), which outputs signals (a function) on group $p4$

$$[\psi \star x](\mathbf{t}, r) := \sum_{\mathbf{p} \in \mathbb{Z}^2} \psi((\mathbf{t}, r)^{-1}\mathbf{p}) \, x(\mathbf{p}). \tag{10}$$

A group $G$ has a natural action on the functions over its elements; if $x : G \to \mathbb{R}$ and $g \in G$, the function $g.x$ is defined as $[g.x](h) := x(g^{-1} \cdot h)$.

---

[2]The definition of group convolution needs to assume that (1) signals $\mathcal{X}(\Omega)$ are in a Hilbert space (to define an inner product $\langle x, \theta \rangle = \int_\Omega x(u)\theta(u)\mathrm{d}u$) and (2) the group $G$ is locally compact (so a Haar measure exists and "shift" of filter can be defined).

[2]Technically, we still need to solve the linear equivariance constraint in Eq. 34 to enable weight-sharing for equivariance, while Weiler and Cesa (2021) have implemented it for 2D case.

For example: The group action of a rotation $r \in C_4$ on the space of functions over $p4$ is

$$[r.y](\mathbf{p}, s) := y(r^{-1}(\mathbf{p}, s)) = y(r^{-1}\mathbf{p}, r^{-1}s), \tag{11}$$

where $r^{-1}\mathbf{p}$ spatially rotates the pixels, $r^{-1}s$ cyclically permutes the 4 channels.

The $G$-space (functions over $p4$) with a natural action of $p4$ on it:

$$[(\mathbf{t}, r).y](\mathbf{p}, s) := y((\mathbf{t}, r)^{-1} \cdot (\mathbf{p}, s)) = y(r^{-1}(\mathbf{p} - \mathbf{t}), r^{-1}s) \tag{12}$$

The group convolution in discrete case is defined as

$$[\psi \star x](g) := \sum_{h \in H} \psi(g^{-1} \cdot h) \, x(h). \tag{13}$$

The group convolution with filter $\psi$ and signal $x$ on $p4$ group is given by:

$$[\psi \star x](\mathbf{t}, r) := \sum_{s \in C_4} \sum_{\mathbf{p} \in \mathbb{Z}^2} \psi((\mathbf{t}, r)^{-1}(\mathbf{p}, s)) \, x(\mathbf{p}, s). \tag{14}$$

Using the fact

$$\psi((\mathbf{t}, r)^{-1}(\mathbf{p}, s)) = \psi(r^{-1}(\mathbf{p} - \mathbf{t}, s)) = [r.\psi](\mathbf{p} - \mathbf{t}, s), \tag{15}$$

the convolution can be equivalently written into

$$[\psi \star x](\mathbf{t}, r) := \sum_{s \in C_4} \left( \sum_{\mathbf{p} \in \mathbb{Z}^2} [r.\psi](\mathbf{p} - \mathbf{t}, s) \, x(\mathbf{p}, s) \right). \tag{16}$$

So $\left( \sum_{\mathbf{p} \in \mathbb{Z}^2} [r.\psi](\mathbf{p} - \mathbf{t}, s) \, x(\mathbf{p}, s) \right)$ can be implemented in usual shift-equivariant convolution CONV2D.

The inner sum $\sum_{\mathbf{p} \in \mathbb{Z}^2}$ is equivalently for the sum in steerable convolution, and the outer sum $\sum_{s \in C_4}$ implement rotation-equivariant convolution that satisfies $H$-steerability kernel constraint. Here, the outer sum is essentially using the *regular* fiber representation of $C_4$.

In other words, group convolution on $p4 = \mathbb{Z}^2 \rtimes C_4$ group is equivalent to steerable convolution on base space $\mathbb{Z}^2$ with the fiber group of $C_4$ with regular representation.

**Stack of feature fields.** Analogous to ordinary CNNs, a feature space in steerable CNNs can consist of multiple feature fields $f_i : \mathbb{Z}^2 \to \mathbb{R}^{c_i}$. The feature fields are stacked $f = \bigoplus_i f_i$ together by concatenating the individual feature fields $f_i$ (along the fiber channel), which transforms under the directly sum $\rho = \bigoplus_i \rho_i$ of individual (fiber) representations. Every layer will be equivariant between input and output field $f_{\text{in}}, f_{\text{out}}$ under induced representations $\pi_{\text{in}}, \pi_{\text{out}}$. For a steerable convolution between more than one-dimensional feature fields, the kernel is matrix-valued (Cohen et al., 2020; Weiler and Cesa, 2021).

## D  SYMMETRIC PLANNING IN PRACTICE

### D.1  BUILDING SYMMETRIC VIN

In this section, we discuss how to achieve Symmetric Planning on 2D grids with $E(2)$-steerable CNNs (Weiler and Cesa, 2021). We focus on implementing symmetric version of value iteration, SymVIN, and generalize the methodology to make a symmetric version of a popular follow-up of VIN, GPPN (Lee et al., 2018).

**Steerable value iteration.** We have showed that, value iteration for path planning problems on $\mathbb{Z}^2$ consists of equivariant maps between steerable feature fields. It can be implemented as an equivariant steerable CNN, with recursively applying two alternating (equivariant) layers:

$$Q_k^a(s) = R_m^a(s) + \gamma \times [P_\theta^a \star V_k](s), \quad V_{k+1}(s) = \max_a Q_k^a(s), \quad s \in \mathbb{Z}^2, \tag{17}$$

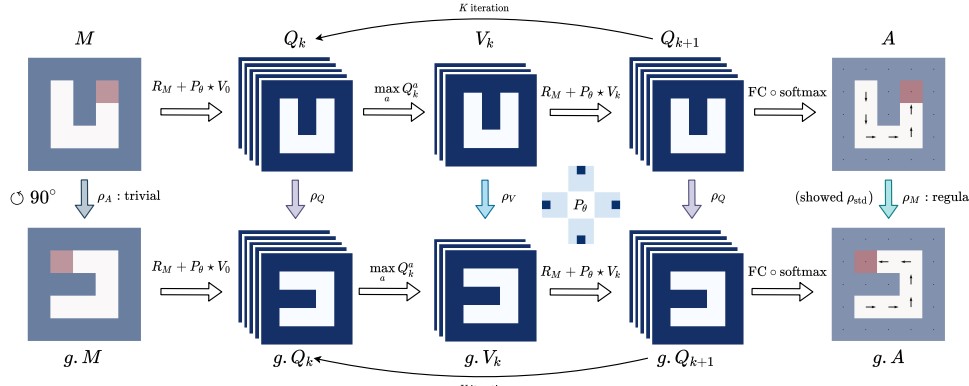

Figure 10: Commutative diagram for the full pipeline of SymVIN on steerable feature fields over $\mathbb{Z}^2$ (every grid). If rotating the input map $M$ by $\pi_M(g)$ of any $g$, the output action $A = \texttt{SymVIN}(M)$ is guaranteed to be transformed by $\pi_A(g)$, i.e. the entire steerable SymVIN is equivariant under induced representations $\pi_M$ and $\pi_A$: $\texttt{SymVIN}(\pi_M(g)M) = \pi_A(g)\texttt{SymVIN}(M)$. We use stacked feature fields to emphasize that SymVIN supports direct-sum of representations beyond scalar-valued.

where $k \in [K]$ indexes iteration, $V_k, Q_k^a, R_m^a$ are steerable feature fields over $\mathbb{Z}^2$ output by equivariant layers, $P_\theta^a$ is a learned kernel in neural network, and $+, \times$ are element-wise operations.

**Implementation of pipeline.** We follow the pipeline in VIN (Tamar et al., 2016a). The commutative diagram for the full pipeline is shown in Figure 10. The path planning task is given by a $m \times m$ spatial binary obstacle occupancy map and one-hot goal map, represented as a feature field $M$ : $\mathbb{Z}^2 \to \{0, 1\}^2$. For the iterative process $Q_k^a \mapsto V_k \mapsto Q_{k+1}^a$, the reward field $R_M$ is predicted from map $M$ (by a $1 \times 1$ convolution layer) and the value field $V_0$ is initialized as zeros. The network output is (logits of) planned actions for all locations[3], represented as $A : \mathbb{Z}^2 \to \mathbb{R}^{|\mathcal{A}|}$, predicted from the final Q-value field $Q_K$ (by another $1 \times 1$ convolution layer). The number of iterations $K$ and the convolutional kernel size $F$ of $P_\theta^a$ are set based on map size $M$, and the spatial dimension $m \times m$ is kept consistent.

**Building Symmetric Value Iteration Networks.** Given the pipeline of VIN fully on steerable feature fields, we are ready to build equivariant version with E(2)-steerable CNNs (Weiler and Cesa, 2021). The idea is to replace every `Conv2d` with a steerable convolution layer between steerable feature fields, and associate the fields with proper fiber representations $\rho(h)$.

VINs use ordinary CNNs and can choose the size of intermediate feature maps. The design choices in steerable CNNs is the feature fields and fiber representations (or *type*) for every layer (Cohen and Welling, 2016a; Weiler and Cesa, 2021). The main difference[4] in steerable CNNs is that we also need to tell the network how to *transform* every *feature field*, by specifying *fiber representations*, as shown in Figure 10.

**Specification of input map and output action.** We first specify *fiber representations* for the input and output field of the network: map $M$ and action $A$. For input **occupancy map and goal** $M$ : $\mathbb{Z}^2 \to \{0, 1\}^2$, it does not $D_4$ to act on the 2 channels, so we use two copies of trivial representations $\rho_M = \rho_{\mathrm{triv}} \oplus \rho_{\mathrm{triv}}$. For **action**, the final action output $A : \mathbb{Z}^2 \to \mathbb{R}^{|\mathcal{A}|}$ is for logits of four actions $\mathcal{A} = (\texttt{north}, \texttt{west}, \texttt{south}, \texttt{east})$ for every location. If we use $H = C_4$, it naturally acts on the four actions (ordered $\circlearrowleft$) by *cyclically $\circlearrowleft$ permuting* the $\mathbb{R}^4$ channels. However, since the $D_4$ group has 8 elements, we need a *quotient representation*, see (Weiler and Cesa, 2021) and Appendix G.

**Specification of intermediate fields: value and reward.** Then, for the intermediate feature fields: Q-values $Q_k$, state value $V_k$, and reward $R_m$, we are free to choose fiber representations, as well as the width (number of copies). For example, if we want 2 copies of regular representation of $D_4$, the feature field has $2 \times 8 = 16$ channels and the stacked representation is $16 \times 16$ (by direct-sum).

---

[3]Technically, it also includes values or actions for obstacles, since the network needs to learn to approximate the reward $R_M(s, \Delta s) = -\infty$ with enough small reward and avoid obstacles.

For the **Q-value field** $Q_k^a(s)$, we use representation $\rho_Q$ and its size as $C_Q$. We need at least $C_A \geq |\mathcal{A}|$ channels for all actions of $Q(s,a)$ as in VIN and GPPN, then stacked together and denoted as $Q_k \triangleq \bigoplus_a Q_k^a$ with dimension $Q_k : \mathbb{Z}^2 \to \mathbb{R}^{C_Q * C_A}$. Therefore, the representation is direct-sum $\bigoplus \rho_Q$ for $C_A$ copies. The **reward** is implemented similarly as $R_M \triangleq \bigoplus_a R_M^a$ and must have same dimension and representation to add element-wisely. For **state value** field, we denote the choose as fiber representation as $\rho_V$ and its size $C_V$. It has size $V_k : \mathbb{Z}^2 \to \mathbb{R}^{C_V}$ Thus, the steerable kernel is *matrix-valued* with dimension $P_\theta : \mathbb{Z}^2 \to \mathbb{R}^{(C_Q * C_A) \times C_V}$. In practice, we found using *regular representations* for all three works the best. It can be viewed as "augmented" state and is related to group convolution, detailed in Appendix G.

**Other operations.** We now visit the remained (equivariant) operations. (1) The $\max$ **operation in** $Q_k \mapsto V_{k+1}$. While we have showed the $\max$ operation in $V_{k+1}(s) = \max_a Q_k^a(s)$ is equivariant in Theorem E.3, we need to apply max(-pooling) for all actions along the "representation channel" from stacked representations $C_A * C_Q$ to one $C_Q$. More details are in Appendix G.2. (2) The **final output layer** $Q_K \mapsto A$. After the final iteration, the $Q$-value field $Q_k$ is fed into the policy layer with $1 \times 1$ convolution to convert the action logit field $\mathbb{Z}^2 \to \mathbb{R}^{|\mathcal{A}|}$.

**Extended method: Symmetric GPPN.** Gated path planning network (GPPN (Lee et al., 2018)) proposes to use LSTM to alleviate the issue of unstable gradient in VINs. Although it does not strictly follow value iteration, it still follows the spirit of steerable planning. Thus, we first obtained a fully convolutional variant of GPPN from [Redacted for anonymous review], called ConvGPPN. It replaces the MLPs in the original LSTM cell with convolutional layers, and then replaces convolutions with equivariant steerable convolutions, resulting in a fully equivariant SymGPPN. See Appendix G.1 for details.

**Extended tasks: planning on learned maps with mapper networks.** We consider two planning tasks on 2D grids: 2D navigation and 2-DOF manipulation. To demonstrate the ability of handling symmetry in differentiable planning, we consider more complicated state space input: visual navigation and workspace manipulation, and discuss how to use mapper networks to convert the state input and use end-to-end learned maps, as in (Lee et al., 2018; Chaplot et al., 2021). See Appendix H.2 for details.

### D.2    PYTORCH-STYLE PSEUDOCODE

Here, we write a section on explaining the SymVIN method with PyTorch-style pseudocode, since it directly corresponds to what we propose in the method section. We try to relate (1) existing concepts with VIN, (2) what we propose in Section 4 and 5 for SymVIN, and (3) actual PyTorch implementation of VIN and SymVIN aligned line-by-line based on semantic correspondence.

We provide the key Python code snippets to demonstrate how easy it is to implement SymVIN, our symmetric version of VIN (Tamar et al., 2016a).

In the current Section 5 (SymPlan practice), we heavily use the concepts from Steerable CNNs. Thanks to the equivariant network community and the `e2cnn` package, the actual implementation is compact and closely corresponds to their non-equivariant counterpart, VIN, line-by-line. Thus, the ultimate goal here is to illustrate that, whatever concepts we have in regular CNNs (e.g., have whatever channels we want), we can can use steerable CNNs that incorporate desired extra symmetry (of $D_4$ rotation+reflection or $C_4$ rotation).

We highlight the implementation of the value iteration procedure in VIN and SymVIN:

$$V := \max_a R^a + \gamma \times P^a * V. \tag{18}$$

Note that we use actual code snippets to avoid hiding any details.

**Defining (steerable) convolution layer.**    First, we show the definition of the key convolution layer for a key operation in VIN and SymVIN: expected value operator, in Listing 1 and 2.

As proved in Theorem E.2, the expected value operator can be executed by a steerable convolution layer for (2D) path planning. This serves as the theoretical foundation on how we should use a steerable layer here.

```
1  import torch
2
3
4
5
6
7
8
9
10
11
12 # Define regular 2D convolution
13 q_conv = torch.nn.Conv2d(
14     in_channels=1,
15     out_channels=2 * q_size,
16     kernel_size=F, stride=1, bias=False
17 )
```

Listing 1: Define 'expected value' convolution layer for VIN.

```
1  import torch
2  import e2cnn
3
4  # Define the symmetry group to be D4
5  gspace = e2cnn.gspaces.FlipRot2dOnR2(N=4)
6  # Define feature (fiber) representations
7  field_type_q_in = e2cnn.nn.FieldType(
8      gspace=gspace,
9      representations=2 * q_size * [gspace.
        regular_repr]
10 )
11 # Define steerable convolution
12 q_r2conv = e2cnn.nn.R2Conv(
13     in_type=field_type_q_in,
14     out_type=field_type_q_out,
15     kernel_size=F, stride=1, bias=False
16 )
```

Listing 2: Define 'expected value' (steerable) convolution layer for SymVIN.

For the left side, a regular 2D convolution is defined for VIN. The right side defines a steerable convolution layer, using the library `e2cnn` from (Weiler and Cesa, 2021). It provides high-level abstraction for building equivariant 2D steerable convolution networks. As a user, we only need to specify how the feature fields transform (as shown in Figure 10), and it will solve the $G$-steerability constraints, process what needs to be trained for equivariant layers, etc. We use name `q_r2conv` to highlight the difference.

**Value iteration procedure.** Second, we compare the for loop for value iteration updates in VIN and SymVIN, where the former one has regular 2D convolution `Conv2D` (Listing 3), and the latter one uses steerable convolution (Weiler and Cesa, 2021) (Listing 4).

The lines are aligned based on semantic correspondence. The `e2cnn` layers, including steerable convolution layers, operate on its `GeometricTensor` data structure, which is to wrap a PyTorch tensor. We denote them with `_geo` suffix. It only additionally needs to specify how this tensor (feature field) transforms under a group (e.g., $D_4$), i.e. the user needs to specify a group representation for it.

`tensor_directsum` is used to concatenate two `GeometricTensor`'s (feature fields) and compute their associated representations (by direct-sum).

Thus, the `e2cnn` steerable convolution layer on the right side `q_r2conv` can be used as a regular PyTorch layer, while the input and output are `GeometricTensor`.

We also define the `max` operation as a customized max-pooling layer, named `q_max_pool`. The implementation is similar to the left side of VIN and needs to additionally guarantee equivariance, and the detail is omitted.

Note that for readability, we assume we use regular representations for the Q-value field $Q$ and the state-value field $V$. They are empirically found to work the best. This corresponds to the definition in `field_type_q_in` in line 9 in the SymVIN definition listing and the comments in line 16-17 in the steerable VI procedure listing for SymVIN.

Other components are omitted.

# E  SYMMETRIC PLANNING FRAMEWORK

This section formulates the notion of Symmetric Planning (SymPlan). We expand the understanding of path planning in neural networks by planning as convolution on steerable feature fields (*steerable planning*). We use that to build *steerable value iteration* and show it is equivariant.

```
1  # Input: maze and goal map, #iterations K
2
3
4
5
6  x = torch.cat([maze_map, goal_map], dim=1)
7
8  r = r_conv(x)
9
10 # Init value function V
11 v = torch.zeros(r.size())
12
13
14 for _ in range(K):
15     # Concat and convolve V with P
16     rv = torch.cat([r, v], dim=1)
17     q = q_conv(rv)
18
19     # Max over action channel
20     # > Q: batch_size x q_size x W x H
21     # > V: batch_size x 1 x W x H
22     q = q.view(-1, q_size, W, H)
23     v, _ = torch.max(q, dim=1)
24     v = v.view(-1, W, H)
25
26 # Output: 'q' (to produce policy map)
```

Listing 3: The central value iteration procedure for VIN. Some variable names are adjusted accordingly for readability. W and H are width and height for 2D map.

```
1  # Input: maze and goal map, #iterations K
2
3  from e2cnn.nn import GeometricTensor
4  from e2cnn.nn import tensor_directsum
5
6  x = torch.cat([maze_map, goal_map], dim=1)
7  x_geo = GeometricTensor(x, type=field_type_x)
8  r_geo = r_r2conv(x_geo)
9
10 # Init V and wrap V in e2cnn 'geometric tensor'
11 v_raw = torch.zeros(r_geo.size())
12 v_geo = GeometricTensor(v_raw, field_type_v)
13
14 for _ in range(K):
15     # Concat (direct-sum) and convolve V with P
16     rv_geo = tensor_directsum([r_geo, v_geo])
17     q_geo = q_r2conv(rv_geo)
18
19     # Max over group channel
20     # > Q: batch_size x (|G| * q_size) x W x H
21     # > V: batch_size x (|G| * 1) x W x H
22     v_geo = q_max_pool(q_geo)
23
24
25
26 # Output: 'q_geo' (to produce policy map)
```

Listing 4: The equivariant steerable value iteration procedure for SymVIN. Lines are aligned by semantic correspondence. Definition of other field types are similar and thus omitted.

### E.1 STEERABLE PLANNING: PLANNING ON STEERABLE FEATURE FIELDS

We start the discussion based on Value Iteration Networks (VINs, (Tamar et al., 2016a)) and use a running example of planning on the 2D grid $\mathbb{Z}^2$. We aim to understand (1) how VIN-style networks embed planning and how its idea generalizes, (2) how is symmetry structure defined in path planning and how could it be injected into such planning networks.

**Constructing $G$-invariant transition: spatial MDP.** Intuitively, the embedded MDP in a VIN is different from the original path planning problem, since (planar) convolutions are translation equivariant but there are different obstacles in different regions.

We found the key insight in VINs is that it implicitly uses an MDP that has translation equivariance. The core idea behind the construction is that it converts *obstacles* (encoded in transition probability $P$, by *blocking*) into "*traps*" (encoded in reward $\bar{R}$, by $-\infty$ *reward*). This allows to use planar convolutions with translation equivariance, and also enables use to further use steerable convolutions.

The demonstration of the idea is shown in **Figure 8 (Left)**. We call it *spatial MDP*, with different transition and reward function $\bar{\mathcal{M}} = \langle \mathcal{S}, \mathcal{A}, \bar{P}, \bar{R}_m, \gamma \rangle$, which converts the "complexity" in the transition function $P$ in $\mathcal{M}$ to the reward function $\bar{R}_m$ in $\bar{\mathcal{M}}$. The state and action space are kept the same: state $\mathcal{S} = \mathbb{Z}^2$ and action $\mathcal{A} \subset \mathbb{Z}^2$ to move $\Delta s$ in four directions in a 2D grid. We provide the detailed construction of the spatial MDP in Section E.3.

**Steerable features fields.** We generalize the idea from VIN, by viewing functions (in RL and planning) as *steerable feature fields*, motivated by (Bronstein et al., 2021a; Cohen et al., 2020; Cohen and Welling, 2016a). This is analogous to pixels on images $\Omega \to [255]^3$, and would allow us to apply convolution on it. The state value function is expressed as a field $V : \mathcal{S} \to \mathbb{R}$, while the $Q$-value function needs a field with $|\mathcal{A}|$ channels: $Q : \mathcal{S} \to \mathbb{R}^{|\mathcal{A}|}$. Similarly, a policy field[5] has probability logits of selecting $|\mathcal{A}|$ actions. For the transition probability $P(s'|s, a)$, we can use action to index it as $P^a(s'|s)$, similarly for reward $R^a(s)$. The next section will show that we can convert the transition function to field and even convolutional filter.

---

[5]We avoid the symbol $\pi$ for policy since it is used for induced representation in (Cohen and Welling, 2016a; Weiler and Cesa, 2021).

### E.2 SYMMETRIC PLANNING: INTEGRATING SYMMETRY BY CONVOLUTION

The seemingly slight change in the construction of spatial MDPs brings important symmetry structure. The general idea in exploiting symmetry in path planning is to use *equivariance* to avoid explicitly constructing equivalence classes of symmetric states. To this end, we construct value iteration over steerable feature fields, and show it is *equivariant* for path planning.

In VIN, the convolution is over 2D grid $\mathbb{Z}^2$, which is symmetric under $D_4$ (rotations and reflections). However, we also know that VIN is already equivariant under translations. To consider all symmetries, as in (Cohen and Welling, 2016a; Weiler and Cesa, 2021), we understand the group $p4m = G = B \rtimes H$ as constructed by a *base space* $B = G/H = (\mathbb{Z}^2, +)$ and a *fiber* group $H = D_4$, which is a *stabilizer subgroup* that fixes the origin $\mathbf{0} \in \mathbb{Z}^2$. We could then formally study such symmetry in the spatial MDP, since we construct it to ensure that the transition probability function in $\bar{\mathcal{M}}$ is $G$-invariant. Specifically, we can uniquely decompose any $g \in \mathbb{Z}^2 \rtimes D_4$ as $t \in \mathbb{Z}^2$ and $r \in D_4$ (and translations act "trivially" on action), so

$$\bar{P}(s' \mid s, a) = \bar{P}(g.s' \mid g.s, g.a) \equiv \bar{P}\left((tr).s' \mid (tr).s, r.a\right), \quad \forall g = tr \in \mathbb{Z}^2 \rtimes D_4, \forall s, a, s'. \quad (19)$$

**Expected value operator as steerable convolution.** The equivariance property can be shown step-by-step: (1) *expected value operation*, (2) *Bellman operator*, and (3) full *value iteration*. First, we use $G$-invariance to prove that the expected value operator $\sum_{s'} P(s'|s,a)V(s')$ is equivariant.

**Theorem E.1.** *If transition is $G$-invariant, the expected value operator $E$ over $\mathbb{Z}^2$ is $G$-equivariant.*

The proof is in Section F.1 and visual understanding is in Figure 8 middle. However, this provides intuition but is inadequate since we do not know: (1) how to implement it with CNNs, (2) how to use multiple feature channels like VINs, since it shows for scalar-valued transition probability and value function (corresponding to trivial representation). To this end, we next prove that we can implement value iteration using steerable convolution with general steerable kernels.

**Theorem E.2.** *If transition is $G$-invariant, there exists a (one-argument, isotropic) matrix-valued steerable kernel $P^a(s - s')$ (for every action), such that the expected value operator can be written as a steerable convolution and is $G$-equivariant:*

$$E^a[V] = P^a \star V, \quad [g.[P^a \star V]](s) = [P^{g.a} \star [g.V]](s), \quad \forall s \in \mathbb{Z}^2, \forall g \in \mathbb{Z}^2 \rtimes D_4. \quad (20)$$

The full derivation is provided in Section F. We write the transition probability as $P^a(s, s')$, and we show it only depends on *state difference* $P^a(s - s')$ (or *one-argument* kernel (Cohen et al., 2020)) using $G$-invariance, which is the key step to show it is some *convolution*. Note that we use one kernel $P^a$ for each action (four directions), and when the group acts on $E$, it also acts on the action $P^{g.a}$ (and state, so technically acting on $\mathcal{S} \times \mathcal{A}$). Additionally, if the steerable kernel also satisfies the $D_4$-*steerability constraint* (Weiler and Cesa, 2021; Weiler et al., 2018), the steerable convolution is *equivariant* under $p4m = \mathbb{Z}^2 \rtimes D_4$. We can then extend VINs from $\mathbb{Z}^2$ translation equivariance to $p4m$-equivariance (translations, rotations, reflections). The derivation follows the existing work on steerable CNNs (Cohen and Welling, 2016b;a; Weiler and Cesa, 2021; Cohen et al., 2020), while this is our goal: to justify the close connection between path planning and steerable convolutions.

**Steerable Bellman operator and value iteration.** We can now represent all operations in Bellman (optimality) operator on steerable feature fields over $\mathbb{Z}^2$ (or *steerable Bellman operator*) as follows:

$$V_{k+1}(s) = \max_a R^a(s) + \gamma \times [P^a \star V_k](s), \quad (21)$$

where $V, R^a, \bar{P}^a$ are steerable feature fields over $\mathbb{Z}^2$. As for the operations, $\max_a$ is (max) pooling (over group channel), $+, \times$ are point-wise operations, and $\star$ is convolution. As the second step, the main idea is to prove every operation in Bellman (optimality) operator on steerable fields is equivariant, including the nonlinear $\max_a$ operator and $+, \times$. Then, iteratively applying Bellman operator forms value iteration and is also equivariant, as shown below and proved in Appendix F.4.

**Proposition E.3.** *For a spatial MDP with $G$-invariant transition, the optimal value function can be found through $G$-steerable value iteration.*

**Remark.** Our framework generalizes the idea behind VINs and enables us to understand its applicability and restrictions. More importantly, this allows us to integrate symmetry but avoid explicitly

building equivalence classes and enables planning with symmetry in end-to-end fashion. We emphasize that the *symmetry in spatial MDPs* is different from *symmetric MDPs* (Zinkevich and Balch, 2001; Ravindran and Barto, 2004; van der Pol et al., 2020a), since our reward function is *not G-invariant* (if not conditioning on reward). Although we focus on $\mathbb{Z}^2$, we can generalize to path planning on higher-dimensional or even continuous Euclidean spaces (like $\mathbb{R}^3$ space (Weiler et al., 2018) or spatial graphs in $\mathbb{R}^3$ (Brandstetter et al., 2021)), and use *equivariant operations* on *steerable feature fields* (such as steerable convolutions, pooling, and point-wise non-linearities) from steerable CNNs. We refer the readers to (Cohen and Welling, 2016b;a; Cohen, 2021; Weiler and Cesa, 2021) for more details.

### E.3 DETAILS: CONSTRUCTING PATH PLANNING IN NEURAL NETWORKS

We provide the detailed construction of doing path planning in neural networks in the Section E. This further explains the visualization in Figure 8 left.

We use the running example of planning on the 2D grid $\mathbb{Z}^2$. We aim to understand (1) how VIN-style networks embed planning and how its idea generalizes, (2) how is symmetry structure defined in path planning and how could it be injected into such planning networks. Recall that we aim to understand (1) how VIN-style networks embed planning and how its idea generalizes, (2) how is symmetry structure defined in path planning and how could it be injected into such planning networks.

**Path planning as MDPs.** To answer the above two questions, we first need to understand how a VIN embeds a path planning problem into a convolutional network as some embedded MDP. Intuitively, the embedded MDP in a VIN is different from the original path planning problem, since (planar) convolutions are translation equivariant but there are different obstacles in different regions.

For path planning on the 2D grid $\mathcal{S} = \mathbb{Z}^2$, the objective is to avoid some obstacle region $\mathcal{C}_{\text{obs}} \subset \mathbb{Z}^2$ and navigate to the goal region $\mathcal{C}_{\text{goal}}$ through free space $\mathcal{C} \backslash \mathcal{C}_{\text{obs}}$. An action $a = \Delta s \in \mathcal{A}$ is to move from the current state $s$ to a next *free* state $s' = s + \Delta s$, where for now we limit it to be in four directions: $\mathcal{A} =$. Assuming deterministic transition, the agent moves to $s'$ with probability 1 if $s + \Delta s \in \mathcal{C} \backslash \mathcal{C}_{\text{obs}}$. If it hits an obstacle, it stays at $s$ if $s + \Delta s \in \mathcal{C}_{\text{obs}}$: $P(s + \Delta s \mid s, \Delta s) = 0$ and $P(s \mid s, \Delta s) = 1$. Every move has a constant negative reward $R(s, a) = -1$ to encourage shortest path. We call this *ground* path planning MDP, a 5-tuple $\mathcal{M} = \langle \mathcal{S}, \mathcal{A}, P, R, \gamma \rangle$.

**Constructing embedded MDPs.** However, such transition function is not translation-invariant, i.e. at different position, the transition probabilities are not related by any symmetry: $P(s'|s, a) \neq P(g.s'|g.s, g.a)$. Instead, we could always construct a "symmetric" MDP that has equivalent optimal value and policy for path planning problems, which is implicitly realized in VINs. The idea is to move the information of obstacles from transition function to reward function: when we hit some action $s + \Delta s \in \mathcal{C}_{\text{obs}}$, we instead allow transition $\bar{P}(s + \Delta s \mid s, \Delta s) = 1$ (with all other $s'$ as 0 probability) while set a "trap" with negative infinity reward $\bar{R}_m(s, \Delta s) = -\infty$. The reward function needs the information from the occupancy map $M$, indicating obstacles $\mathcal{C}_{\text{obs}}$ and free space. For the free region, the reward is still a constant $\bar{R}_M(s, \Delta s) = -1$, indicating the cost of movement.

We call it the *embedded* MDP, with different transition and reward function $\bar{\mathcal{M}} = \langle \mathcal{S}, \mathcal{A}, \bar{P}, \bar{R}_M, \gamma \rangle$, which converts the "complexity" in the transition function $P$ in $\mathcal{M}$ to the reward function $\bar{R}_m$ in $\bar{\mathcal{M}}$. Here, map $M$ shall also be treated as an "input", thus later we will derive how the group acts on the map $g.M$. It has the same optimal policy and value as the ground MDP $\mathcal{M}$, since the optimal policies in both MDPs will avoid obstacles in $\mathcal{M}$ or trap cells in $\bar{\mathcal{M}}$. It could be easily verified by simulating value iteration backward in time from the goal position.

The transition probability $\bar{P}$ of the embedded MDP $\bar{\mathcal{M}}$ is for an "empty" maze and thus translation-invariant. Note that the reward function $\bar{R}$ is not not necessarily invariant. This construction is not limited to 2D grid and generalizes to continuous state space or even higher dimensional space, such as $\mathbb{R}^6$ configuration space for 6-DOF manipulation.

Note, all of this is what we use to conceptually understand how a VIN is possible to learn. The reward cannot be negative infinity, but the network will learn it to be smaller than all desired Q-values.

### E.4 DETAILS: UNDERSTANDING SYMMETRIC PLANNING BY ABSTRACTION

How do we deal with potential symmetry in path planning? How do we characterize it? We try to understand symmetric planning (steerable planning after integrating symmetry with equivariance) and how it is difference classic planning algorithms, such as A*, for planning under *symmetry*.

**Steerable planning.** Recall that we generalize the idea of VIN by considering it as a planning network that composes of mappings between steerable feature fields.

The critical point is that, convolutions directly operate on local patches of pixels and never directly touch coordinates of pixels. In analogy, this avoids a critical drawback in other *explicit* planning algorithms: in sampling-based planning, a trajectory $(s_1, a_1, s_2, a_2, \ldots)$ is sampled and inevitable represented by states $\Omega = \mathcal{S}$. However, to find another symmetric state $g.s$, we potentially need to compare it against all known states $\mathcal{S}' \subset \mathcal{S}$ with all symmetries $g \in G$. On high level, an implicit planner can avoid such symmetry breaking and is more easily compatible with symmetry by using equivariant constraints.

We can use MDP homomorphism to understand this (Ravindran and Barto, 2004; van der Pol et al., 2020b).

**MDP homomorphisms.** An *MDP homomorphism* $h : \mathcal{M} \to \overline{\mathcal{M}}$ is a mapping from one MDP $\mathcal{M} = \langle \mathcal{S}, \mathcal{A}, P, R, \gamma \rangle$ to another $\overline{\mathcal{M}} = \langle \overline{\mathcal{S}}, \overline{\mathcal{A}}, \overline{P}, \overline{R}, \gamma \rangle$ (Ravindran and Barto, 2004; van der Pol et al., 2020b). $h$ consists of a tuple of surjective maps $h = \langle \phi, \{\alpha_s \mid s \in \mathcal{S}\} \rangle$, where $\phi : \mathcal{S} \to \overline{\mathcal{S}}$ is the state mapping and $\alpha_s : \mathcal{A} \to \overline{\mathcal{A}}$ is the *state-dependent* action mapping. The mappings are constructed to satisfy the following conditions:

$$\overline{R}\left(\phi(s), \alpha_s(a)\right) \triangleq R(s, a) ,$$
$$\overline{P}\left(\phi\left(s'\right) \mid \phi(s), \alpha_s(a)\right) \triangleq \sum_{s'' \in \phi^{-1}(\phi(s'))} P\left(s'' \mid s, a\right) , \tag{22}$$

for all $s, s' \in \mathcal{S}$ and for all $a \in \mathcal{A}$.

We call the *reduced* MDP $\overline{\mathcal{M}}$ the *homomorphic image* of $\mathcal{M}$ under $h$. If $h = \langle \phi, \{\alpha_s \mid s \in \mathcal{S}\} \rangle$ has *bijective* maps $\phi$ and $\{\alpha_s\}$, we call $h$ an *MDP isomorphism*. Given MDP homomorphism $h$, $(s, a)$ and $(s', a')$ are said to be $h$-equivariant if $\sigma(s) = \sigma(s')$ and $\alpha_s(a) = \alpha_{s'}(a')$.

**Symmetry-induced MDP homomorphisms.** Given group $G$, an MDP homomorphism $h$ is said to be *group structured* if any state-action pair $(s, a)$ and its transformed counterpart $g.(s, a)$ are mapped to the same abstract state-action pair: $(\phi(s), \alpha_s(a)) = (\phi(g.s), \alpha_{g.s}(g.a))$, for all $s \in \mathcal{S}, a \in \mathcal{A}, g \in G$. For convenience, we denote $g.(s, a)$ as $(g.s, g.a)$, where $g.a$ implicitly[6] depends on state $s$. Applied to the transition and reward functions, the transition function $P$ is $G$-invariant if $P$ satisfies $P(g.s'|g.s, g.a) = P(s'|s, a)$, and reward function $R$ is $G$-invariant if $R(g.s, g.a) = R(s, a)$, for all $s \in \mathcal{S}, a \in \mathcal{A}, g \in G$.

However, this only fits the type of symmetry in (van der Pol et al., 2020a; Wang et al., 2021). And also, they cannot handle invariance to translation $\mathbb{Z}^2$. In our case, we need to augment the reward function with map $M$ input:

$$R_{g.M}(g.s, g.a) = R_M(s, a), \tag{23}$$

for all $s \in \mathcal{S}, a \in \mathcal{A}, g \in G = p4m$.

This means that, at least for rotations and reflections $D_4$, the MDPs constructed from transformed maps $\{g.M\}$ are MDP *isomorphic* to each other.

---

[5]We avoid the symbol $\pi$ for policy since it is used for induced representation in (Cohen and Welling, 2016a; Weiler and Cesa, 2021).

[6]The group operation acting on action space $\mathcal{A}$ *depends on state*, since $G$ actually acts on the *product space* $\mathcal{S} \times \mathcal{A}$: $(g, (s, a)) \mapsto g.(s, a)$, while we denote it as $(g.s, g.a)$ for consistency with $h = \langle \phi, \{\alpha_s \mid s \in \mathcal{S}\} \rangle$. As a bibliographical note, in van der Pol et al. (2020b), the group acting on state and action space is denoted as state transformation $L_g : \mathcal{S} \to \mathcal{S}$ and *state-dependent* action transformation $K_g^s : \mathcal{A} \to \mathcal{A}$.

### E.5 NOTE: AUGMENTED STATE

We derive the relationship between group convolution and steerable convolution in Section C.4.

The augmented state $\mathbb{Z}^2 \rtimes D_4 \to \mathbb{R}$ can be similarly treated on the group $p4m = \mathbb{Z}^2 \rtimes D_4$. It is equivalent to using regular representation on the base space $\mathbb{Z}^2$ as $\mathbb{Z}^2 \to \mathbb{R}^8$.

## F SYMMETRIC PLANNING FRAMEWORK: PROOFS

We show the derivation and proofs for all theoretical results in this section.

We follow the notation in (Cohen et al., 2020) to use $\star$ for (one-argument) convolution and $\cdot$ for (two-argument) multiplication:

$$E^a[V](s) = [P^a \cdot V](s) \equiv \sum_{s'} P^a\left(s' \mid s\right) \cdot V(s') \tag{24}$$

### F.1 PROOF: EQUIVARIANCE OF SCALAR-VALUED EXPECTED VALUE OPERATION

We present the Theorem E.1 here and its formal definition.

**Theorem F.1.** *If transition is $G$-invariant, the expected value operator $E$ over $\mathbb{Z}^2$ is $G$-equivariant:*

$$[g.E^a[V]]\,(s) = [E^{g.a}[g.V]]\,(s), \quad \text{for all } g = tr \in \mathbb{Z}^2 \rtimes D_4.$$

*Proof.* $E$ is the expected value operator. We also write the transition probability as

Recall the $G$-invariance condition of transition probability, the group element $g$ acts on $s, a, s'$:

$$\bar{P}(s' \mid s, a) = \bar{P}(g.s' \mid g.s, g.a) \equiv \bar{P}\left((tr).s' \mid (tr).s, r.a\right), \quad \forall g = tr \in \mathbb{Z}^2 \rtimes D_4, \forall s, a, s', \tag{25}$$

where we can uniquely decompose any $g \in \mathbb{Z}^2 \rtimes D_4$ as $t \in \mathbb{Z}^2$ and $r \in D_4$ (Cohen and Welling, 2016a). Note that, since the action is the difference between states $a = \Delta s = s' - s$, the translation part $t$ acts trivially on it, so $g.a = (tr).a = r.a$ for all $r \in D_4$.

We transform the feature field and show its equivariance:

$$[g.E^a[V]](s) \equiv [g.[P^a \cdot V](s) \tag{26}$$

$$\equiv \sum_{s'} \rho_{\text{triv}}(r) P^a\left(s' \mid (tr)^{-1}.s\right) \cdot V(s') \tag{27}$$

$$= \sum_{s'} \rho_{\text{triv}}(r) P^{r.a}\left((tr).s' \mid s\right) \cdot V(s') \tag{28}$$

$$= \sum_{\tilde{s}'} \rho_{\text{triv}}(r) P^{r.a}\left(\tilde{s}' \mid s\right) \cdot V\left((tr)^{-1}\tilde{s}'\right) \tag{29}$$

$$= \sum_{\tilde{s}'} P^{r.a}\left(\tilde{s}' \mid s\right) \cdot \rho_{\text{triv}}(r) V\left((tr)^{-1}\tilde{s}'\right) \tag{30}$$

$$\equiv [P^{r.a} \cdot [g.V]](s) \tag{31}$$

$$\equiv [E^{r.a}[g.V]](s). \tag{32}$$

We use the trivial representation $\rho_{\text{triv}}(g) = \text{Id}_{1\times1} = 1$ to emphasize that (1) the group element $g$ acts on *feature fields* $P^a$ and $V$, and (2) both feature fields $P^a$ and $V$ are scalar-valued and correspond to the one-dimensional trivial representation of $r \in D_4$.

In the third line, we use the $G$-invariance of transition probability.

The fourth line uses substitution $\tilde{s}' \triangleq (tr).s'$, for all $s' \in \mathbb{Z}^2$ and $tr \in \mathbb{Z}^2 \rtimes D_4$. This is an one-to-one mapping and the summation does does not change.

□

F.2  PROOF: *expected value operator* AS STEERABLE CONVOLUTION

In this section, we derive how to cast expected value operator as steerable convolution. The equivariance proof is in the next section.

In Theorem E.1, we show equivariance of value iteration in 2D path planning, while it is only for the case that feature fields $P^a$ and $V$ are scalar-valued and correspond to one-dimensional trivial representation of $r \in D_4$.

Here, we provide the derivation for Theorem E.2 show that steerable CNNs (Cohen and Welling, 2016a) can achieve value iteration since we could construct the G-invariant transition probability as a steerable convolutional kernel. This generalizes Theorem E.1 from scalar-valued kernel (for transition probability) with trivial representation to matrix-valued kernel with any combination of representations, enabling using stack (direct-sum) of feature fields and representations.

We state Theorem E.2 here for completeness:

**Theorem F.2.** *If transition is $G$-invariant, there exists a (one-argument, isotropic) matrix-valued steerable kernel $P^a(s - s')$ (for every action), such that the expected value operator can be written as a steerable convolution and is $G$-equivariant:*

$$E^a[V] = P^a \star V, \quad [g.[P^a \star V]](s) = [P^{g.a} \star [g.V]](s), \quad \forall s \in \mathbb{Z}^2, \forall g \in \mathbb{Z}^2 \rtimes D_4. \tag{33}$$

**Steerable kernels.**  In our earlier definition, $\psi^a$ and $f_{\text{in}}$ are transition probability and value function, which are both real-valued $\psi^a : \mathbb{Z}^2 \to \mathbb{R}, f_{\text{in}} : \mathbb{Z}^2 \to \mathbb{R}$. However, this is a *special case* which corresponds to use one-dimensional *trivial representation* of the fiber group $D_4$. In the general case in steerable CNNs (Cohen and Welling, 2016a; Weiler and Cesa, 2021), we can choose the feature fields $\psi^a : \mathbb{Z}^2 \to \mathbb{R}^{C_{\text{out}} \times C_{\text{in}}}$ and $f_{\text{in}} : \mathbb{Z}^2 \to \mathbb{R}^{C_{\text{in}}}$ and their fiber representations, which we will introduce the group representations of $D_4$ and how to choose in practice in the next section.

Weiler et al. (2018) show that *convolutions* with *steerable kernels* $\psi^a : \mathbb{Z}^2 \to \mathbb{R}^{C_{\text{out}} \times C_{\text{in}}}$ is the most general *equivariant linear map* between steerable feature space, transforming under $\rho_{\text{in}}$ and $\rho_{\text{out}}$. In analogy to the continuous version[7] in (Weiler and Cesa, 2021), the convolution is equivariant *iff* the kernel satisfies a $H$-steerability kernel constraint:

$$\psi^a(hs) = \rho_{\text{out}}(h)\psi^a(s)\rho_{\text{in}}(h^{-1}) \quad h \in H = D_4, s \in \mathbb{Z}^2. \tag{34}$$

**Expected value operation as steerable convolution.**  The foremost step is to show that the expected value operation is a form of convolution and is also $G$-equivariant. By definition, if we want to write a (linear) operator as a form of convolution, we need one-argument kernel. Cohen et al. (2020) show that every linear equivariant operator is some convolution and provide more details. For our case, this is formally shown as follows.

**Proposition F.3.** *If the transition probability is $G$-invariant, it can be expressed as an (one-argument) kernel $P^a(s'|s) = P^a(s' - s)$ that only depends on the difference $s' - s$.*

*Proof.* The form of our proof is similar to (Cohen et al., 2020), while its direction is different from us. We construct a MDP such that the transition probability kernel is $G$-invariant, while Cohen et al. (2020) assume the linear operator $\psi \cdot f$ is linear *equivariant* operator on a homogeneous space, and then derive that the kernel is $G$-invariant and expressible as one-argument kernel. Additionally, our kernel $\psi^a(s, s')$ and $\psi^a(s - s')$ both live on the base space $B = \mathbb{Z}^2$ but not on the group $G = \mathbb{Z}^2 \rtimes D_4$.

We show that the transition probability only depends on the difference $\Delta s = s' - s$, so we can define the two-argument kernel $P^a(s'|s)$ on $\mathcal{S} \times \mathcal{S}$ by an one-argument kernel $P^a(s' - s)$ (for every action

---

[7]Weiler and Cesa (2021) use letter $G$ to denote the stabilizer subgroup $H \leq \mathrm{O}(2)$ of $\mathrm{E}(2)$.

$a$) on $\mathcal{S} = \mathbb{Z}^2$, without loss of generality:

$$P^a(s' - s) \equiv P^a(\mathbf{0}, s' - s) \tag{35}$$

$$= P^{g.a}(g.\mathbf{0}, g.(s' - s)) \tag{36}$$

$$= P^{r.a}((rs).\mathbf{0}, (rs).(s' - s)) \tag{37}$$

$$= P^{r.a}(r.s, r.(s' - s + s)) \tag{38}$$

$$= P^{r.a}(r.s, r.s') \tag{39}$$

$$= P^a(s, s'), \tag{40}$$

where the second step uses $G$-invariance with $g = sr$, understood as the composition of a translation $s \in \mathbb{Z}^2$ and a transformation in $r \in D_4$.

$\square$

Additionally, we can also derive that, for the one-argument kernel, if we rotate state difference $r.(s' - s)$, the probability is the same for rotated action $r.a$.

$$P^a(s' - s) = P^{r.a}(r.(s' - s)), \text{for all } r \in D_4, s, s' \in \mathbb{Z}^2 \tag{41}$$

The *expected value operator* with two-argument kernel can be then written as

$$E[V](s) \equiv [P^a \cdot V](s) = \sum_{s'} P^a(s'|s)V(s') = \sum_{s'} P^a(s' - s)V(s') \equiv [P^a \star V](s). \tag{42}$$

Note that we do not differentiate between cross-correlation ($s' - s$) and convolution ($s - s'$).

### F.3 PROOF: EQUIVARIANCE OF *expected future value*

Our derivation follows the existing work on group convolution and steerable convolution networks (Cohen and Welling, 2016b;a; Weiler and Cesa, 2021; Cohen et al., 2020). However, the goal of providing the proof is not just for completeness, but instead to emphasize the close connection between how we formulate our planning problem and the literature of steerable CNNs, which explains and justifies our formulation.

Additionally, there are several subtle differences worth to mention. (1) Throughout the paper, we do not discuss kernels or fields that live on a group $G$ to make it more approachable. Nevertheless, group convolutions are a special case of steerable convolutions with fiber representation $\rho$ as regular representation. (2) We use $\mathbb{Z}^2$ as running example. Some prior work uses $\mathbb{R}^2$ or $\mathbb{Z}^2$, but they are merely just differ in integral and summation. (3) The definition of convolution and cross-correlation might be defined and used interchangeably in the literature of (equivariant) CNNs.

**Notation.** To keep notation clear and consistent with the literature (Cohen and Welling, 2016a; Cohen et al., 2020; Weiler and Cesa, 2021), we denote the transition probability $\bar{P}(s'|s, a) \triangleq \psi^a(s, s') \in \mathbb{R}$ (one kernel for an action) and value function as $V(s') \triangleq f_{\text{in}}(s') \in \mathbb{R}$, and the resulting expected value as $f_{\text{out}}^a(s) = \sum_{s'} \psi^a(s, s') f_{\text{in}}(s')$ (given a specific action $a$).

**Transformation laws: induced representation.** For some group acting on the base space $\mathbb{Z}^2$, the signals $f : \mathbb{Z}^2 \to \mathbb{R}^c$ are transformed like Cohen and Welling (2016a):

$$[\pi(g)f](x) = f(g^{-1}x) \tag{43}$$

Apply a translation $t$ and a transformation $r \in D_4$ to $f$, we get $\pi(tr)f$. The transformation law on the input space $f_{\text{in}}$ is (Cohen and Welling, 2016a; Weiler and Cesa, 2021):

$$f(x) \mapsto [\pi(tr)f](x) \triangleq \rho(r) \cdot \left[f\left((tr)^{-1}x\right)\right] \tag{44}$$

The transformation law of the output space after applying $\pi_{\text{in}}$ on input $f_{\text{in}}$ is given by Cohen and Welling (2016a):

$$[\psi \star f](x) \mapsto [\psi \star [\pi(tr)f]](x) \triangleq \rho(r) \cdot \left[[\psi \star f]\left((tr)^{-1}x\right)\right]. \tag{45}$$

In our case, the output space is $f_{\text{out}}^a : \mathbb{Z}^2 \to \mathbb{R}^{C_{\text{out}}}$ and the input space is $f_{\text{in}} : \mathbb{Z}^2 \to \mathbb{R}^{C_{\text{in}}}$. Intuitively, if we rotate a vector field (fibers represent arrows) by the induced representation $\pi(tr)$ of $f$, we also need to rotate the direction of arrows by $\rho(r), r \in D_4$.

**Equivariance.** Now we prove the steerable convolution is equivariant:

$$[\psi^a \star [\pi_{\text{in}}(g) f_{\text{in}}]] (s) = [\pi_{\text{out}}(g) f_{\text{out}}^a] (s) \quad \forall s \in \mathcal{S}, \forall g \in G. \tag{46}$$

The induced representation of input field $f_{\text{in}}$ is induced by the fiber representation $\rho_{\text{in}}$, expressed by $\pi_{\text{in}} \triangleq \text{ind}_H^G \rho_{\text{in}} = \text{ind}_{D_4}^{\mathbb{Z}^2 \rtimes D_4} \rho_{\text{in}}$, where $\rho_{\text{in}}$ is the fiber representation of group $H = D_4$. The induced representation of output field $\pi_{\text{out}}$ is analogously from $\rho_{\text{out}}$.

Weiler and Cesa (2021) proved equivariance of steerable convolutions for $\mathbb{R}^2$ case, while we include the proof under our setup for completeness. The definition in (Weiler and Cesa, 2021) uses a form of *cross-correlation* and we use *convolution*, while it is usually referred to interchangeably in the literature and is equivalent. Cohen and Welling (2016a); Weiler et al. (2018); Weiler and Cesa (2021); Cohen et al. (2020); Cohen (2021) provide more details and we refer the readers to them for more comprehensive account.

The convolution on discrete grids $\mathbb{Z}^2$ with input field $f_{\text{in}}$ transformed by the induced representation $\pi_{\text{in}}$ gives:

$$
\begin{aligned}
[\psi^a \star [\pi_{\text{in}}(rt) f_{\text{in}}]](s) &= \sum_{s' \in \mathbb{Z}^2} \psi^a(s - s')[\pi_{\text{in}}(rt) f_{\text{in}}](s') \\
&= \sum_{s' \in \mathbb{Z}^2} \psi^a(s - s') \rho_{\text{in}}(r) f_{\text{in}}(r^{-1}(s' - t)) \\
&= \sum_{s' \in \mathbb{Z}^2} \rho_{\text{out}}(r) \psi^a(r^{-1}(s - s')) \rho_{\text{in}}(r)^{-1} \rho_{\text{in}}(r) f_{\text{in}}(r^{-1}(s' - t)) \\
&= \rho_{\text{out}}(r) \sum_{s' \in \mathbb{Z}^2} \psi^a(r^{-1}(s - s')) f_{\text{in}}(r^{-1}(s' - t)) \\
&= \rho_{\text{out}}(r) \sum_{\tilde{s} \in \mathbb{Z}^2} \psi^a(r^{-1}(s - t) - \tilde{s}) f_{\text{in}}(\tilde{s}) \\
&= \rho_{\text{out}}(r) f_{\text{out}}(r^{-1}(s - t)) \\
&= [\pi_{\text{out}}(rt) f_{\text{out}}^a] (s),
\end{aligned} \tag{47}
$$

where $s' \in \mathcal{S} = \mathbb{Z}^2$, and thus satisfies the equivariance condition:

$$[\psi^a \star [\pi_{\text{in}}(rt) f_{\text{in}}]] (s) = [\pi_{\text{out}}(rt) f_{\text{out}}^a] (s), \forall s \in \mathbb{Z}^2, \forall rt \in \mathbb{Z}^2 \rtimes D_4. \tag{48}$$

1. Definition of $\star$

2. Transformation law of the induced representation $\pi_{\text{in}}$ (Cohen and Welling, 2016a; Weiler and Cesa, 2021)

3. Kernel steerability $\psi^a(s) = \rho_{\text{out}}(h) \psi^a(h^{-1}s) \rho_{\text{in}}(h^{-1})$ (Weiler and Cesa, 2021)

4. Move and cancel

5. Substitutes $\tilde{s} = r^{-1}(s' - t)$, $r^{-1}s' = r^{-1}t + \tilde{s}$, so $r^{-1}(s - s') = r^{-1}(s - t) - \tilde{s}$. Since $r \in D_4$ and $s - s' \in \mathbb{Z}^2$, the result is still in $p4m$, it is one-to-one correspondence $p4m \times \mathbb{Z}^2 \to \mathbb{Z}^2$, and the summation does not change. Weiler and Cesa (2021) analogously considers the continuous case, where $D_4$ is orthogonal transformations so the Jacobian is always 1.

6. Definition of $\star$

7. Transform law of the induced representation $\pi_{\text{out}}$

### F.4 Proof: equivariance of steerable value iteration

As the third and final step, we would like to show that the full steerable value iteration pipeline is equivariant under $G = \mathbb{Z}^2 \rtimes D_4$. We need to show that every operation in the steerable value iteration is equivariant.

The key is to prove that $\max_a$ is an equivariant non-linearity over feature fields, which follows Section D.2 in (Weiler and Cesa, 2021).

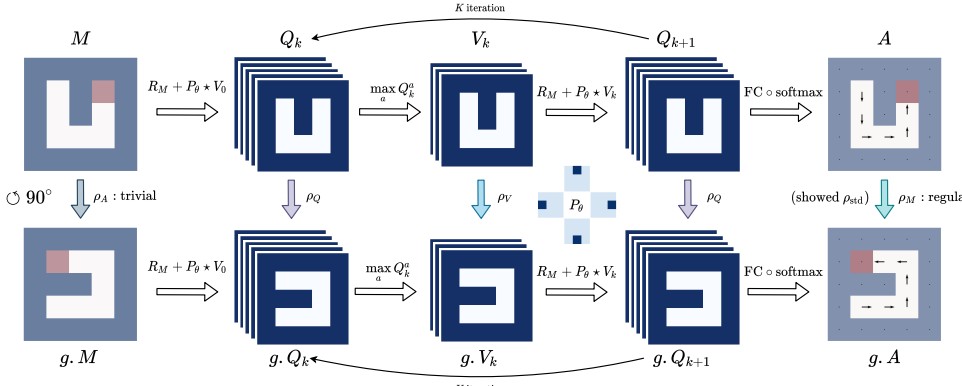

Figure 11: *We attach a copy of the commutative diagram of SymVIN to show the equivariance of steerable value iteration.* Commutative diagram for the full pipeline of SymVIN on steerable feature fields over $\mathbb{Z}^2$ (every grid). If rotating the input map $M$ by $\pi_M(g)$ of any $g$, the output action $A = \texttt{SymVIN}(M)$ is guaranteed to be transformed by $\pi_A(g)$, i.e. the entire steerable SymVIN is equivariant under induced representations $\pi_M$ and $\pi_A$: $\texttt{SymVIN}(\pi_M(g)M) = \pi_A(g)\texttt{SymVIN}(M)$. We use stacked feature fields to emphasize that SymVIN supports direct-sum of representations beyond scalar-valued.

**Step 1: $V \mapsto Q$.** Here, we prove the equivariance of $Q_k^a(s) = \bar{R}_M^a(s) + \gamma \times \left[\bar{P}_\theta^a \star V_k\right](s)$. First, let the group acts on both sides:

$$Q_k^a(s) = \bar{R}_M^a(s) + \gamma \times \left[\bar{P}_\theta^a \star V_k\right](s) \tag{49}$$

$$\Longleftrightarrow [\pi_{\text{out}}(g)Q_k^a](s) = [\pi_{\text{out}}(g)\bar{R}_M^a](s) + \gamma \times \left[\pi_{\text{out}}(g)\left[\bar{P}_\theta^a \star V_k\right]\right](s) \tag{50}$$

$$\Longleftrightarrow [\pi_{\text{out}}(g)Q_k^a](s) = [\pi_{\text{out}}(g)\bar{R}_M^a](s) + \gamma \times \left[\bar{P}_\theta^a \star [\pi_{\text{in}}(g)V_k]\right](s) \tag{51}$$

$$\Longleftrightarrow Q_k^{g.a}(g^{-1}s) = \bar{R}_{g.M}^{g.a}(g^{-1}s) + \gamma \times \left[\bar{P}_\theta^{g.a} \star V_k\right](g^{-1}s) \tag{52}$$

$$\Longleftrightarrow Q_k^{\tilde{a}}(\tilde{s}) = \bar{R}_{\pi_M(g)M}^{\tilde{a}}(\tilde{s}) + \gamma \times \left[\bar{P}_\theta^{\tilde{a}} \star V_k\right](\tilde{s}) \tag{53}$$

The the last step we substitute $\tilde{s} = g^{-1}s$ and $\tilde{a} = g.a$.

$M : \mathbb{Z}^2 \to \{0,1\}^2$ is the concatenation of maze occupancy map and goal map, which also lives on $\mathbb{Z}^2$. We use two copies of trivial representations as fiber representation $\rho_M$, and denote the induced representation of the field $M$ as $\pi_M$.

Then, we prove the equivariance: if we transform the occupancy map (and goal map), the value iteration should have both input $V$ and output $Q$ transformed. Since this is an iterative process, the only input to the value iteration is actually the occupancy map $M : \mathbb{Z}^2 \to \{0,1\}^2$.

Before that, we observe that the reward also has $G$-invariance when we have map as input:

$$\bar{R}_M^a(s) = \bar{R}_{g.M}^{g.a}(g.s). \tag{54}$$

Additionally, since the reward $\bar{R}_M^a(s)$ means the reward at given position in map $M$ **after executing action** $a$, when we transform the map, we also need to transform the action: $\bar{R}_{g.M}^{g.a}(s)$.

Since it is iterative process, let the $Q$-map being transformed by $g$:

$$[g.Q_k^a](s) = Q_k^a(g^{-1}s) \tag{55}$$

$$= \bar{R}_M^a(g^{-1}s) + \gamma \times \left[\bar{P}_\theta^a \star V_k\right](g^{-1}s) \tag{56}$$

$$= \bar{R}_{g.M}^{g.a}(s) + \gamma \times \left[\bar{P}_\theta^a \star V_k\right](g^{-1}s) \tag{57}$$

$$= \bar{R}_{g.M}^{g.a}(s) + \gamma \times \left[\bar{P}_\theta^{g.a} \star [g.V_k]\right](s) \tag{58}$$

The second last step uses the $G$-invariance condition $\bar{R}_M^a(s) = \bar{R}_{g.M}^{g.a}(g.s)$. The last step uses the equivariance of steerable convolution.

It should be understood as: (1) transforming map $g.M$ and action $g.a$, is always equal to (2) transforming values $[g.Q_k^a]$ and $[g.V_k]$. This proves the equivariance visually shown in Figure 11.

**Step 2: $Q \mapsto V$.** The second step is to show for $V_{k+1}(s) = \max_a Q_k^a(s)$.

Intuitively, we sum over every channel of each representation. For example, if we have $N$ copies of the regular representation with size $|D_4| = 8$, we transform the tensor $(N \times 8) \times m \times m$ to $(1 \times 8) \times m \times m$ along the $N$ channel. Thus, how we use the $8 \times 8$ regular representation to transform the $N \times 8$ channels still holds for $1 \times 8$, which implies equivariance. The $m \times m$ spatial map channels form the base space $\mathbb{Z}^2$ and are transformed as usual (spatially rotated).

Weiler and Cesa (2021) provide detailed illustration and proofs for equivariance of different types of non-linearities.

**Step 3: multiple iterations.** Since each layer is equivariant (under induced representations), Cohen and Welling (2016b); Kondor and Trivedi (2018); Cohen et al. (2020) show that stacking multiple equivariant layers is also equivariant. Thus, we know iteratively applying step 1 and 2 (*equivariant steerable Bellman operator*) is also *equivariant* (*steerable value iteration*).

# G Implementation Details

## G.1 Implementation of SymGPPN

ConvGPPN (Kong, 2022) is inspired by VIN and GPPN. To avoid the training issues in VIN, GPPN proposes to use LSTM to alleviate them. In particular, it does not use max pooling in the VIN. Instead, it uses a CNN and LSTM to mimic the value iteration process. ConvGPPN, on the other hand, integrates CNN into LSTM, resulting in a single component convLSTM for value iteration. We found that ConvGPPN performs better than GPPN in most cases. Based on ConvGPPN, SymGPPN replaces each convolutional layer with steerable convolutional layer.

## G.2 Implementation of max operation

Here, we consider how to implement the $\max$ operation in $V_{k+1}(s) = \max_a Q_k^a(s)$. The $\max$ is taken over every state, so the computation mainly depends on our choice of fiber representation.

For example, if we use *trivial representations* for both input and output, the input would be $Q_k : \mathbb{Z}^2 \to \mathbb{R}^{1*C_A}$ and the output is state-value $V_k : \mathbb{Z}^2 \to \mathbb{R}$. This recovers the default value iteration since we take $\max$ over $\mathbb{R}^{C_A}$ vector.

In steerable CNNs, we can use stack of fiber representations. We can choose from regular-regular, trivial-trivial, and regular-trivial (trivial-regular is not considered).

We already covered *trivial* representations for both input and output, they would be $Q_k : \mathbb{Z}^2 \to \mathbb{R}^{C_Q*C_A}$ and $V_k : \mathbb{Z}^2 \to \mathbb{R}^{C_V}$ with $C_Q = C_V = 1$, since every channel would need a trivial representation.

If we use *regular* representation for $Q$ and *trivial* for $V$, they are $Q_k : \mathbb{Z}^2 \to \mathbb{R}^{C_Q*C_A}$ and $V_k : \mathbb{Z}^2 \to \mathbb{R}^{C_V}$ with $C_Q = |D_4| = 8$ and $C_V = 1$. It degenerates that we just take $\max$ over all $C_Q * C_A$ channels.

For both using regular representations, we need to make sure they use the same fiber group (such as $D_4$ or $C_4$), so $C_Q = C_V$. If using $D_4$, we have $Q_k : \mathbb{Z}^2 \to \mathbb{R}^{8*C_A}$ and $V_k : \mathbb{Z}^2 \to \mathbb{R}^8$, and we take $\max$ over every $C_A$ channels (for every location) and have 8 channels left, which are used as $\mathbb{Z}^2 \to \mathbb{R}^8$.

Empirically, we found using regular representations for both works the best overall.

# H Experiment Details and Additional Results

## H.1 Environment Setup

**Action space.** Note that the MDP action space $\mathcal{A}$ needs to be *compatible* with the group action $G \times \mathcal{A} \to \mathcal{A}$. Since the E2CNN package (Weiler and Cesa, 2021) uses *counterclockwise* rotations as generators for rotation groups $C_n$, the action space needs to be *counterclockwise*.

We show the figures for **Configuration-space and Workspace manipulation** in Figure 12, and the figures for **2D and Visual Navigation** in Figure 13.

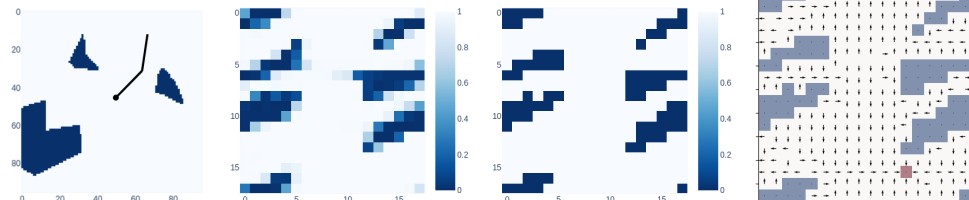

Figure 12: A set of visualization for a 2-joint manipulation task. The obstacles are randomly generated. **(1)** The 2-joint manipulation task shown in top-down workspace with $96 \times 96$ resolution. This is used as the input to the **Workspace Manipulation** task. **(2)** The predicted configuration space in resolution $18 \times 18$ from a mapper module, which is jointly optimized with a planner network. **(3)** The ground truth configuration space from a handcraft algorithm in resolution $18 \times 18$. This is used as input to the **Configuration-space (C-space) Manipulation** task and as target in the auxiliary loss for the Workspace Manipulation task (as done in SPT (Chaplot et al., 2021)). **(4)** The predicted policy (overlaid with C-space obstacle for visualization) from an end-to-end trained SymVIN model that uses a mapper to take the top-down workspace image and plans on a learned map. The red block is the goal position.

**Manipulation.** For planning in configuration space, the configuration space of the 2 DoFs manipulator has no constraints in the $\{0, \pi\}$ boundaries, i.e., no joint limits. To reflect this nature of the configuration space in manipulation tasks, we use circular padding before convolution operation. The circular padding is applied to convolution layers in VIN, SymVIN, ConvGPPN, and SymGPPN. Moreover, in GPPN, there is a convolution encoder before the LSTM layer. We add the circular padding in the convolution layers in GPPN as well.

In **2-DOF manipulation** in configuration space, we adopt the setting in (Chaplot et al., 2021) and train networks to take as input of configuration space, represented by two joints. We randomly generate $0$ to $5$ obstacles in the manipulator workspace. Then the 2 degree-of-freedom (DOF) configuration space is constructed from workspace and discretized into 2D grid with sizes $\{18, 36\}$, corresponding to bins of $20°$ and $10°$, respectively.

We allow each joint to rotate over $2\pi$, so the configuration space of 2-DOF manipulation forms a torus $\mathbb{T}^2$. Thus, the both boundaries need to be connected when generating action demonstrations, and (equivariant) convolutions need to be circular (with padding mode) to wrap around for all methods. We allow each joint to rotate over $2\pi$, so the both boundaries in configuration space need to be connected when generating action demonstrations, and (equivariant) convolutions need to be circular (with padding mode) to wrap around for all methods.

### H.2    BUILDING MAPPER NETWORKS

**For visual navigation.** For navigation, we follow the setting in GPPN (Lee et al., 2018). The input is $m \times m$ panoramic egocentric RGB images in 4 directions of resolution $32 \times 32 \times 3$, which forms a tensor of $m \times m \times 4 \times 32 \times 32 \times 3$. A mapper network converts every image into a 256-dimensional embedding and results in a tensor in shape $m \times m \times 4 \times 256$ and then predicts map layout $m \times m \times 1$.

For the first image encoding part, we use a CNN with first layer of 32 filters of size $8 \times 8$ and stride of $4 \times 4$, and second layer with 64 filters of size $4 \times 4$ and stride of $2 \times 2$, with a final linear layer of size 256.

The second obstacle prediction part, the first layer has 64 filters and the second layer has 1 filter, all with filter size $3 \times 3$ and stride $1 \times 1$.

**For workspace manipulation.** For **workspace manipulation**, we use U-net Ronneberger et al. (2015) with residual-connection He et al. (2015) as a mapper, see Figure.14. The input is $96 \times 96$ top-down occupancy grid of the workspace with obstacles, and the target is to output $18 \times 18$ configuration space as the maps for planning.

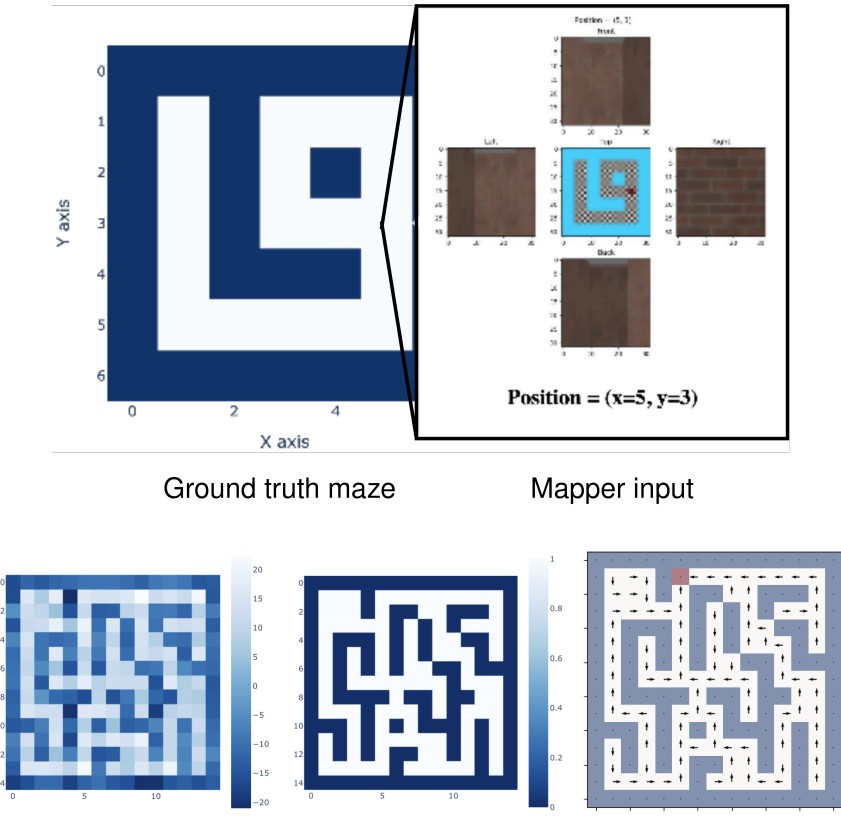

Ground truth maze        Mapper input

Figure 13: A set of visualization for 2D navigation and visual navigation. The maze is randomly generated. **(1, top)** The 3D visual navigation environment generated by an illustrative $7 \times 7$ map, where we highlight the panoramic view at a position $(5, 3)$ with four RGB images (resolution $32 \times 32 \times 3$). The entire observation tensor for this $7 \times 7$ example visual navigation environment is $7 \times 7 \times 4 \times 32 \times 32 \times 3$. This is used as the input to the **Visual Navigation** task. **(2)** Another predicted map in resolution $15 \times 15$ from a mapper module, which is jointly optimized with a planner network. We show the visualization a different map used in actual training. **(3)** The ground truth map in resolution $15 \times 15$. This is also used as input to the **2D Navigation** task and as target in the auxiliary loss for the Visual Navigation task (as done in GPPN). **(4)** The predicted policy from an end-to-end trained SymVIN model that uses a mapper to take the observation images (formed as a tensor) and plans on a learned map. The red block is the goal position.

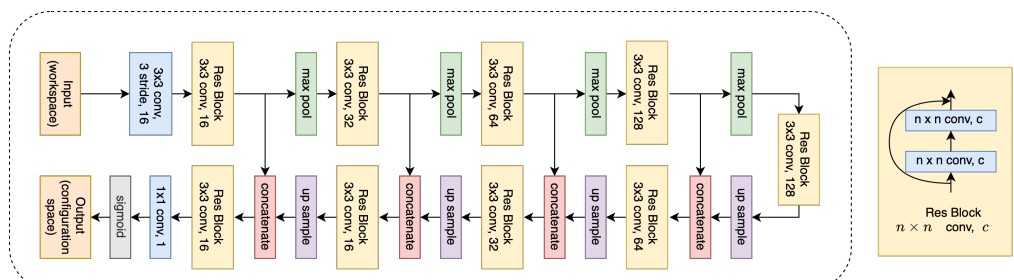

Figure 14: The U-net architecture we used as manipulation mapper.

During training, we pre-train the mapper and the planner separately for 15 epochs. Where the mapper takes manipulator workspace and outputs configuration space. The mapper is trained to minimize the binary cross entropy between output and ground truth configurations space. The planner is trained in the same way as described in Section 6.1. After pre-training, we switch the input to the planner from ground truth configuration space to the one from the mapper. During testing, we follow the pipeline in Chaplot et al. (2021) that the mapper-planner only have access to the manipulator workspace.

### H.3 TRAINING SETUP

We try to mimic the setup in VIN and GPPN (Lee et al., 2018).

For non-SymPlan related parameters, we use learning rate of $10^{-3}$, batch size of 32 if possible (GPPN variants need smaller), RMSprop optimizer.

For SymPlan parameters, we use 150 hidden channels (or 150 *trivial* representations for SymPlan methods) to process the input map. We use 100 hidden channels for Q-value for VIN (or 100 *regular* representations for SymVIN), and use 40 hidden channels for Q-value for GPPN and ConvGPPN (or 40 *regular* representations for SymGPPN on $15 \times 15$, and 20 for larger maps because of memory constraint).

### H.4 VISUALIZATION OF LEARNED MODELS

We visualize a trained VIN and a SymVIN, evaluated on a $15 \times 15$ map and its rotated version. For non-symmetric VIN in Figure 15, the learned policy is obviously not equivariant under rotation.

We also visualize SymVIN on larger map sizes: $28 \times 28$ and $50 \times 50$, to demonstrate its performance and equivariance.

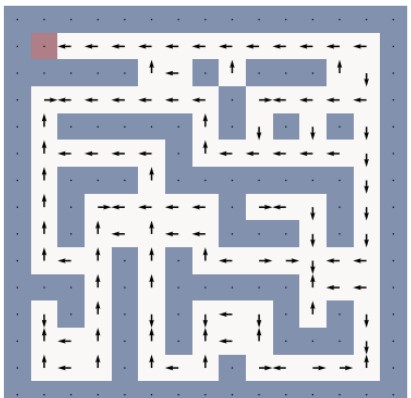
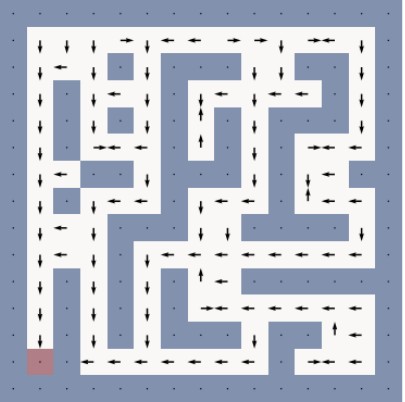

Figure 15: A trained VIN evaluated on a $15 \times 15$ map and its rotated version. It is obvious that the learned policy is not equivariant under rotation.

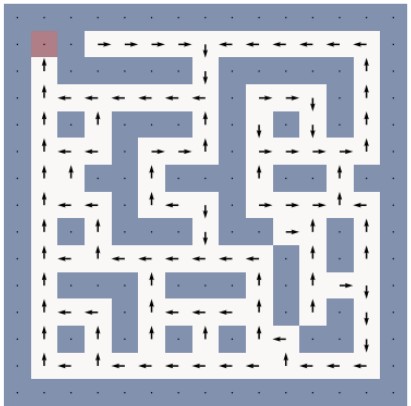
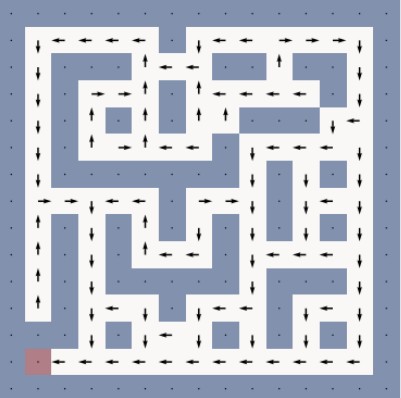

Figure 16: A trained SymVIN evaluated on a $15 \times 15$ map and its rotated version.

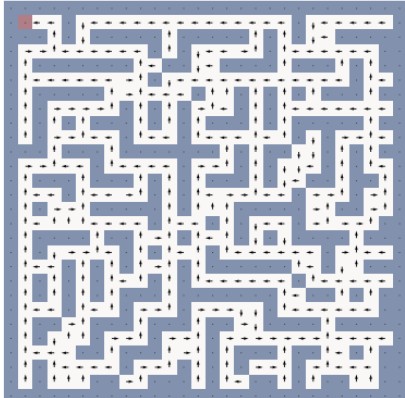 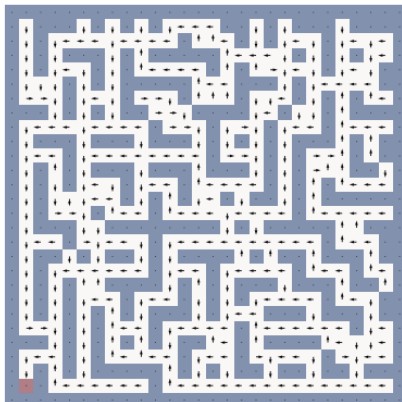

Figure 17: A fully trained SymVIN evaluated on a $28 \times 28$ map and its rotated version.

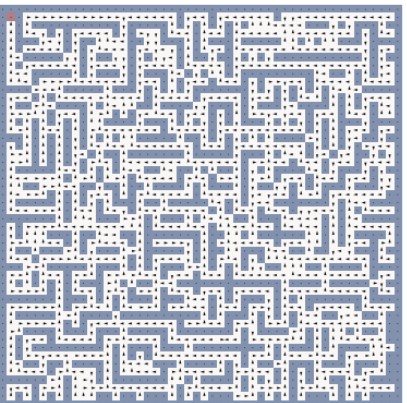 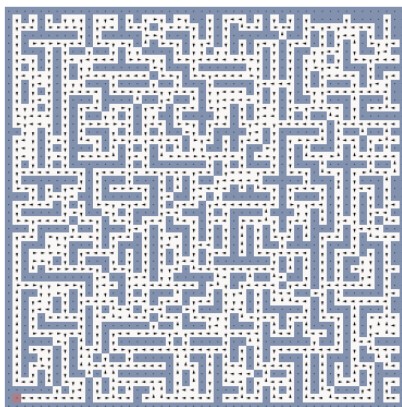

Figure 18: A fully trained SymVIN evaluated on a $50 \times 50$ map and its rotated version.

## H.5   SPL METRIC

We additionally provide SPL metric of the 2D navigation experiments with different seeds. The trends should be similar to the results in the main paper.

Table 2: Averaged test success rate (%) over 5 seeds for using 10K/2K/2K dataset on 2D navigation.

| Methods | $15 \times 15$ | $28 \times 28$ | $50 \times 50$ |
|---|---|---|---|
| VIN | $69.94_{\pm 2.92}$ | $61.43_{\pm 5.65}$ | $54.62_{\pm 9.76}$ |
| SymVIN | $98.94_{\pm 0.39}$ | $97.56_{\pm 0.71}$ | $96.27_{\pm 0.27}$ |
| GPPN | $96.48_{\pm 0.65}$ | $94.35_{\pm 1.14}$ | $88.25_{\pm 4.25}$ |
| ConvGPPN | $99.51_{\pm 0.19}$ | $96.44_{\pm 8.56}$ | $96.62_{\pm 0.38}$ |
| SymGPPN | $99.68_{\pm 0.00}$ | $99.98_{\pm 0.02}$ | $99.91_{\pm 0.05}$ |

The performance under SPL metric is pretty similar to success rate. This indicates that most paths are pretty close to the optimal length from all planners. Thus, we decided to not include this to avoid extra concept in experiment section.

## H.6   ABLATION STUDY

**Additional training curves.**   We also provide other training curves that we only show test numbers in the main text.

Table 3: Averaged test SPL (%) over 5 seeds for using 10K/2K/2K dataset on 2D navigation.

| Methods | $15 \times 15$ | $28 \times 28$ | $50 \times 50$ |
|---------|------------|------------|------------|
| VIN | $68.62_{\pm 3.02}$ | $60.81_{\pm 5.73}$ | $54.14_{\pm 9.90}$ |
| SymVIN | $98.08_{\pm 0.50}$ | $97.23_{\pm 0.67}$ | $96.07_{\pm 0.27}$ |
| GPPN | $95.55_{\pm 0.64}$ | $94.00_{\pm 1.17}$ | $87.79_{\pm 4.30}$ |
| ConvGPPN | $98.73_{\pm 0.19}$ | $96.22_{\pm 8.57}$ | $96.44_{\pm 0.40}$ |
| SymGPPN | $98.91_{\pm 0.01}$ | $99.76_{\pm 0.06}$ | $99.82_{\pm 0.06}$ |

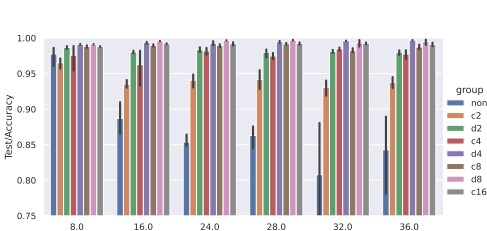 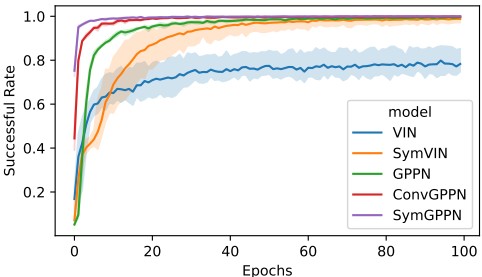

Figure 19: **(Left)** Accuracy evaluated on unseen test maps. The x-axis is the width of the map, and the y-axis is the accuracy, reported on every map size and every size and every chose symmetry group $G$. **(Right)** Visual navigation $15 \times 15$ with 10K data.

**Training efficiency with less data.** Since the supervision is still dense, we experiment on training with even smaller dataset to experiment in more extreme setup. We experiment how symmetry may affect the training efficiency of Symmetric Planners by further reducing the size of training dataset. We compare on two environments: 2D navigation and visual navigation, with training/validation/test size of 1K/200/200, for all methods.

**Choose of symmetry groups for navigation.** One important benefit of partially equivariant network is that, we do not need to design the group representation of MDP action space $\rho_{\mathcal{A}}(g)$ for different group or action space. Thus, we experiment several $G$-equivariant variants with different group equivariance: (discrete rotation group) $C_2, C_4, C_8, C_{16}$, and (dihedral group) $D_2, D_4, D_8$, all based on $E(2)$-steerable CNN (Weiler and Cesa, 2021). For all intermediate layers, we use regular representations $\rho_{\text{reg}}(g)$ of each group, followed by a final policy layer with non-equivariant $1 \times 1$ convolution.

The results are reported in the Figure 19 (left). We only compare VIN (denoted as "none" symmetry) against our $E(2)$-VIN (other symmetry group option) on 2D navigation with $15 \times 15$ maps.

In general, the planners equipped with any $G$ group equivariance outperform the vanilla non-equivariant VIN, and $D_4$-equivariant steerable CNN performs the best on most map sizes. Additionally, since the environment has actions in 8 directions (4 diagonals), $C_8$ or $D_8$ groups seem to take advantage of that and have slightly higher accuracy on some map sizes, while $C_{16}$ is over-constrained compared to the true symmetry $G = D_4$ and be detrimental to performance. The non-equivariant VIN also experiences higher variance on large maps.

**Choosing fiber representations.** As we use steerable convolutions (Weiler and Cesa, 2021) to build symmetric planners, we are free to choose the representations for feature fields, where intermediate equivariant convolutional layers will be equivariant between them $f(\rho_{\text{in}}(g)x) = \rho_{\text{out}}(g)f(x)$. We found representations for some feature fields are critical to the performance: mainly $V : \mathcal{S} \to \mathbb{R}$ and $Q : \mathcal{S} \to \mathbb{R}^{|A|}$.

We use the best setting as default, and ablate every option. As shown in Table 4, changing $\rho_V$ or $\rho_Q$ to trivial representation would result in much worse results.

**Fully vs. Partially equivariance for symmetric planners.** One seemingly minor but critical design choice in our SymPlan networks is the choice of the final policy layer, which maps Q-values $\mathcal{S} \to \mathbb{R}^{|\mathcal{A}|}$ to policy logits $\mathcal{S} \to \mathbb{R}^{|\mathcal{A}|}$. Fully equivariant is expected to perform better, but it has some points worth to mention. (1) We experience unstable training at the beginning, where the loss can

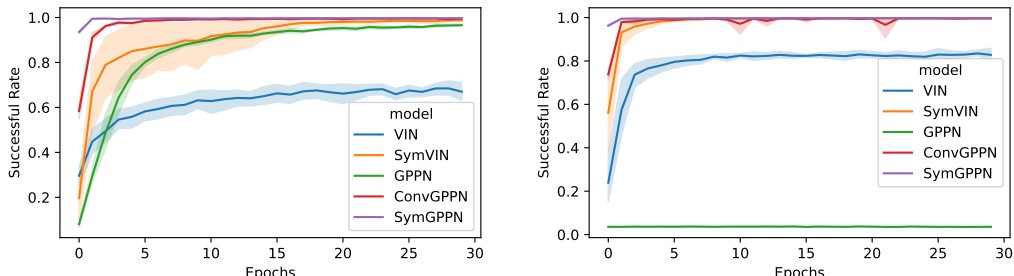

Figure 20: Training curves on **(Left)** 2D navigation with 10K of $15 \times 15$ maps and on **(Right)** 2DoFs manipulation with 10K of $18 \times 18$ maps in configuration space. Faded areas indicate standard error.

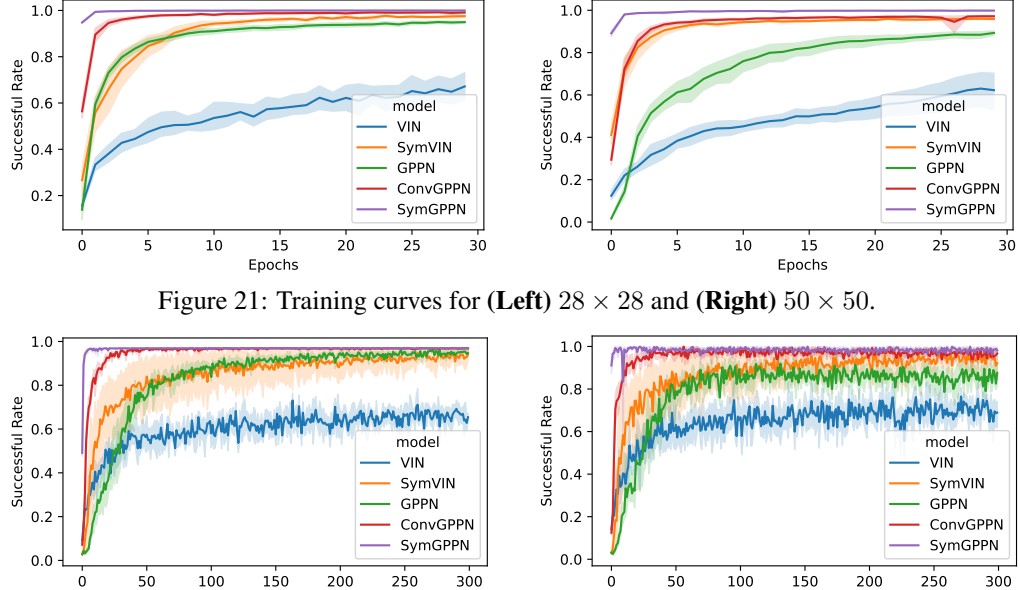

Figure 21: Training curves for **(Left)** $28 \times 28$ and **(Right)** $50 \times 50$.

Figure 22: Training curves for $15 \times 15$ 2D navigation 1K data **(Left)** training and **(Right)** validation successful rate.

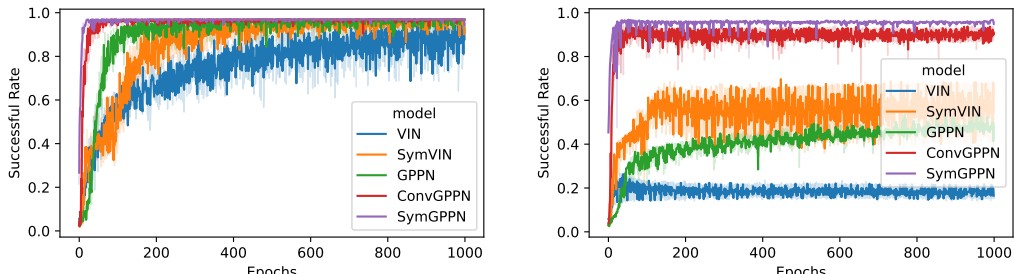

Figure 23: Training curves for $15 \times 15$ visual navigation 1K data **(Left)** training and **(Right)** validation successful rate.

Table 4: Fiber representations

| (Fiber representation) | SymVIN |
|---|---|
| Default | 98.45 |
| Hidden: trivial to regular | 99.07 |
| State-value $\rho_V$: regular to trivial | 63.08 |
| Q-value $\rho_Q$: regular to trivial | 21.30 |
| $\rho_Q$ and $\rho_V$: both trivial | 2.814 |

go up to $10^6$ in the first epoch, while we did not observe it in non-equivariant or partially equivariant counterparts. However, this only slightly affects training.

In summary, we found even though fully equivariant version can perform slightly better in the best tuned setting, on average setting, partially equivariant version is more robust and the gap is much larger, as shown in the follow table, which an example of averaging over three choices of representations introduced in the last paragraph. On average partially equivariant version is much better. In our experiments, partially equivariant version also is easier to tune.

Table 5: Fully vs. Partially equivariance

| (Equivariance) | SymVIN |
|---|---|
| *Partially* equivariant averaged over all representations | 91.04 |
| *Fully* equivariant averaged over all representations | 42.61 |

**Generalization additional experiment for fixed $K$.** For fixed $K$ setup in Figure 24 (left), we keep number of iterations to be $K = 30$ and kernel size $F = 3$ for all methods.

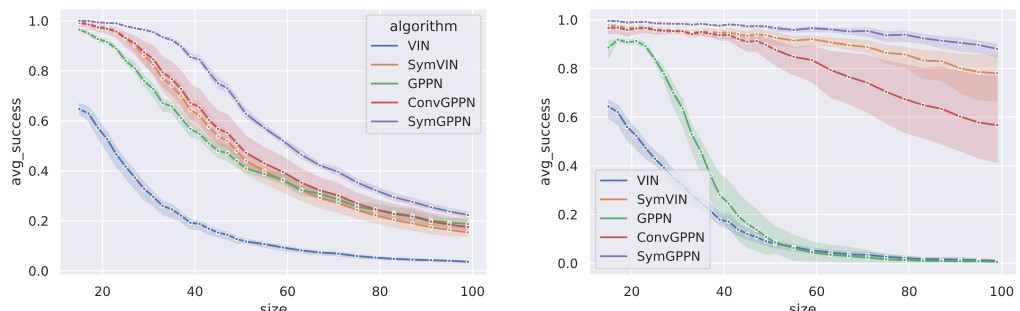

Figure 24: Results for generalization on larger maps for all methods. **(Left)** Fixed $K = 30$ iterations. **(Left)** Variable $K$ iterations, where $K = \sqrt{2} \cdot M$ and $M$ is the generalization map size (x-axis).

For SymVIN, it far surpasses VIN for all sizes and preserves the gap throughout the evaluation. Additionally, SymVIN has slightly higher variance across three random seeds (three separately trained models).

Among GPPN and its variants, SymGPPN significantly outperforms both GPPN and ConvGPPN. Interestingly, ConvGPPN has sharper drop with map size than both SymGPPN and ConvGPPN and thus has increasingly larger gap with SymGPPN and finally even got surpassed by GPPN. Across random seeds, the three trained models of ConvGPPN give unexpectedly high variance compared to GPPN and SymGPPN.

