# OpenReview forum: "Integrating Symmetry into Differentiable Planning with Steerable Convolutions"
_ICLR.cc/2023/Conference — ICLR 2023 poster_

### Official Review · Reviewer_qcZN · 2022-10-25

**Confidence:** 2
**Correctness:** 4
**Technical Novelty And Significance:** 3
**Empirical Novelty And Significance:** 3
**Recommendation:** 8

**Clarity, Quality, Novelty And Reproducibility:**

The presentation of this paper looks clear to me.

The quality of the definitions, proofs, and experiments is also fine.

Although the ideas and techniques are not totally new, finding appropriate connections among those concepts should be appreciated.

I think their method is reproducible. Moreover, they already promised that the code would be open-sourced.


**Strength And Weaknesses:**

This paper flows very well and smoothly. The idea of exploiting the rotation and reflection symmetry in the path finding problem is simple but also elegant. And it is a good choice to start from Z2 space and VIN with steerable convolution. The authors formally build the connections among symmetries, value iterations, and steerable convolutions. Moreover, the experiment results further verify that this symmetry regularization does help.

Besides those strengths, I might have some questions in mind.

First, although it is intuitive that symmetries could improve the performance and the experiments also demonstrate the effectiveness of such proposed inductive bias empirically, is it possible to prove or even compute quantitatively why symmetry can help the path finding problems?

Second, it is also mentioned that the symmetry structure also exists in other higher-dimensional continuous Euclidean spaces or spatial graphs (of path finding problems), can the authors provide more clues on how to generalize their method to those cases? Since the current experiments are on small scales and those mentioned more general spaces could be more practical in some real-world problems.

Third, in Section 6.3, why is iteration K set to be sqrt(2)M? And why is the iteration linear to the map size?


**Summary Of The Paper:**

This paper demonstrates one way to exploit the rotation and reflection symmetry structure in path planning problems. More specifically, the authors use Value Iteration Networks (VINs) to solve the path finding problem on the Z2 space. They show that the steerable convolution can incorporate the symmetry structure in the value iteration process. Experiments show that their proposed symmetry regularization can improve the training speed, training success rate, as well as generalizability.

**Summary Of The Review:**

This paper presents an elegant symmetry regularization to the VIN-based path finding pipeline. The methodology part reads very well and experiments are sufficient to back up their technical novelties. In summary, I think this paper looks good to me.

---

> ### Author Response · Authors · 2022-11-16
> **Response to Reviewer qcZN**
>
> We appreciate the reviewer for the time and effort spent reviewing our work. We address the concerns through individual responses. We are happy to answer more concrete concerns the reviewer might have.
>
> First, although it is intuitive that symmetries could improve the performance and the experiments also demonstrate the effectiveness of such proposed inductive bias empirically, **is it possible to prove or even compute quantitatively why symmetry can help the path finding problems**?
>
> - We did try to show how symmetry helps path planning in Appendix Section E.4 and provide some intuition here:
>     - We show that the MDPs from rotated maps are related by MDP homomorphisms (or more strictly, isomorphisms), which has been used in Ravindran et al. 2004. Intuitively, the states can be related by $D_4$ rotations/reflections are aggregated into one state.
>     - Thus, we inject this symmetry property to the SymVIN algorithm via enforcing $D_4$-invariance of the value function on $\mathbb{Z}^2$, which is equivalent to a function on the quotient space (intuitively, $\mathbb{Z}^2$ quotient out $D_4$).
>     - By adding the global and local equivariance to $D_4$, our Symmetric Planning algorithms can effectively plan on the quotient/reduced MDP with smaller state space. By local equivariance, we mean that every value iteration step is a convolution (or other equivariant) operation and is equivariant to rotation/reflection (see Figure 3 and 4), thus the search space in planning can ben shrink exponentially w.r.t. planning horizon.
>     - Although we haven’t formally prove this or the sample complexity side, but intuitively this gives $|G|$ times smaller state space and sample complexity.
> - More generally, how equivariance helps in general supervised learning tasks is still an open problem. Recent work includes showing better generalization bound of the equivariant networks.
>     - Sannai, Akiyoshi, Masaaki Imaizumi, and Makoto Kawano. "Improved generalization bounds of group invariant/equivariant deep networks via quotient feature spaces." *Uncertainty in Artificial Intelligence*. PMLR, 2021.
>     - Elesedy, Bryn, and Sheheryar Zaidi. "Provably strict generalisation benefit for equivariant models." *International Conference on Machine Learning*. PMLR, 2021.
>     - Behboodi, Arash, Gabriele Cesa, and Taco Cohen. "A PAC-Bayesian Generalization Bound for Equivariant Networks." NeurIPS 2022.
>
> Second, it is also mentioned that **the symmetry structure also exists in other higher-dimensional continuous Euclidean spaces or spatial graphs (of path finding problems), can the authors provide more clues on how to generalize their method to those cases?** Since the **current experiments are on small scales** and those mentioned more **general** **spaces could be more practical in some real-world problems.**
>
> - Yes, we can use the sphere as an example. Suppose the task is to navigate on a 2D sphere $S^2$ embedded into a 3D space. It has equivariance under 3D rotation group $SO(3)$.
>     - Thus, the map can be represented as a signal on the sphere $f: S^2 \to \mathbb{R}$, which is in the vector space $L^2(S^2)$. Thus, we only need to replace the E(2)-steerable CNN with another equivariant network under SO(3) rotations.
> - SO(3)-equivariant networks have been studied in the equivariant network field, such as Cohen et al. (ICLR 2018) titled Spherical CNNs.
>     - It decomposes the signal by generalized Fourier transformation onto irreducible representations of $SO(3)$, which are the spherical harmonics. It then performs group convolutions operations (proven to be equivariant) on $S^2$ and $SO(3)$ and applies inverse Fourier transformation back to the spatial domain.
> - There are also gauge equivariant CNNs (can run convolution on arbitrary manifolds) and equivariant message passing GNNs (run convolution on graphs while being equivariant to e.g. rotations/translations), which can potentially support path planning on more spaces.

---

### Official Review · Reviewer_hMHM · 2022-10-25

**Confidence:** 3
**Correctness:** 4
**Technical Novelty And Significance:** 3
**Empirical Novelty And Significance:** 3
**Recommendation:** 8

**Clarity, Quality, Novelty And Reproducibility:**

Clarity was good, the work appears to be novel and implementation details are provided for reproducibility.

**Strength And Weaknesses:**

# Strengths
* The paper focuses on an important problem and clearly motivates the need for equivariance for path planning (with good illustrative examples).
* The authors theoretically analyze the problem and provide key theorems to support the proposed algorithm
* The experiment setup is sensible and uses good baselines
* The results are strong and demonstrate consistent improvements over the baselines

# Weaknesses
* The results are only demonstrated only small and toyish maps (100 x 100 or smaller). In robotics planning, the maps can be significantly larger (e.g., 960 x 960 in Active Neural SLAM, Chaplot et al., ICLR 2020). It is unclear how the proposed method works on more complex scenarios.
* The metrics in Table 1, Figures 6 and 7 only capture success rate and not efficiency of the predicted plan relative to an optimal planner like A*. A common metric to measure this efficiency is SPL (Success weighted by Path Length) from Anderson et al., 2018: https://arxiv.org/pdf/1807.06757.pdf

**Summary Of The Paper:**

The paper proposes to build differentiable planning methods that are equivariant to translations, rotations, and reflection transforms. This is done by replacing standard convolutional layers (which are equivariant to translations) in differentiable planning algorithms like VINs and GPPNs with steerable convolutional layers (which are additionally equivariant to rotations and reflections). The authors also theoretically show that value operators and value iteration on 2D grids are equivariant, and this can be expressed as a steerable convolution. Experiments are performed on 2D grid / visual navigation and 2 DOF manipulation and demonstrate strong improvements over non-equivariant versions of the planning methods.

**Summary Of The Review:**

# Post-rebuttal update
The authors address the comments about the map sizes and optimality of paths for planning. I'm raising my rating to accept.

# Pre-rebuttal review
The paper tackles an important problem and proposes to incorporate equivariances in differentiable planning methods. It provides both theoretical and empirical analyses and demonstrates strong improvements over prior work. However, the results are demonstrated only in simplistic maps and do not capture closeness of planned paths to the optimal path.

---

> ### Author Response · Authors · 2022-11-16
> **Response to Reviewer hMHM**
>
> We appreciate the reviewer for the time and effort spent reviewing our work. We address the concerns through individual responses and also additional results in the updated paper. We are happy to answer more concrete concerns the reviewer might have, such as "correctness (3: Some of the paper’s claims have minor issues. A few statements are not well-supported, or require small changes to be made correct)" and "technical/empirical novelty (3: The contributions are significant and somewhat new. Aspects of the contributions exist in prior work.)".
>
> The results are **only demonstrated only small and toyish maps** (100 x 100 or smaller). In robotics planning, **the ·maps can be significantly larger (e.g., 960 x 960 in Active Neural SLAM, Chaplot et al., ICLR 2020)**. It is **unclear how the proposed method works on more complex scenarios.**
>
> - In terms of the problem scale:
>     - We choose this because we follow the prior work (as pointed out, in VIN, GPPN, SPT, and so on). Specifically, VIN mainly experimented on 15x15, and GPPN mainly used 15x15 and tried 28x28 as a scalability experiment. SPT is advertised to be much better scalable with Transformers and used up to 50x50.
>     - We think integrating symmetry into differentiable planning is an orthogonal topic with the scalability of differentiable planning algorithms, although symmetry could potentially help with scalability.
>     - Furthermore, we also have the visual navigation experiment, where we train the localization/mapping module end-to-end. Our method can learn from panoramic egocentric RGB images of all locations ($15 \times 15 \times 4 \times 32 \times 32 \times 3$).
>     - Additionally, the generalization experiment shows that the model can generalize to larger maps, which unveils the potential of scalability. This has not been done in prior work along this line.
> - In comparison with Active Neural SLAM:
>     - Active Neural SLAM is more like a systematic pipeline, and our approach aims to improve only the differentiable planning module. Specifically, it uses a hierarchical architecture with the global planner and local planner to support coarse-to-fine structure, so the planning horizon of either planner is not that long. Technically, our Symmetric Planners should serve as an alternative to either global planner or local planner, but not comparing with the whole Active Neural SLAM pipeline.
>     - Neural SLAM uses room-like maps, so it has a different structure (such as more empty space compared to our maze-like maps) and requires less planning horizon.
>
> The metrics in Table 1, Figures 6 and 7 only capture the success rate and not the efficiency of the predicted plan relative to an optimal planner like A*. A common metric to measure this efficiency is SPL (Success weighted by Path Length) from Anderson et al., 2018: [https://arxiv.org/pdf/1807.06757.pdf](https://arxiv.org/pdf/1807.06757.pdf)
>
> - We did use SPL in our experiments. We found that SPL is pretty close to the success rate (like 99 vs 98), which seems to suggest that the paths found by value iteration are mostly close to optimal.
> - Thus, we thought it can bring an extra unnecessary concept, so we decided to use the success rate only for a wider audience and did not use it.
> - **We attach the new SPL results in the updated paper appendix Section B.** The performance under the SPL metric is pretty similar to the success rate. This indicates that most paths are pretty close to the optimal length of all planners.

---

> > ### Comment · Reviewer_hMHM · 2022-11-22
> > **Reviewer response**
> >
> > I thank the authors for addressing my comments. I would like to suggest that the authors add a reference to the SPL results in the main paper, stating how the planned paths are not only successful, but also close to optimal. I'm happy to increase my rating to accept.

---

### Official Review · Reviewer_QPW3 · 2022-11-01

**Confidence:** 3
**Correctness:** 3
**Technical Novelty And Significance:** 2
**Empirical Novelty And Significance:** Not applicable
**Recommendation:** 6

**Clarity, Quality, Novelty And Reproducibility:**

The overall quality is sound, as the motivation for integrating symmetric constraint into VIN (as a linear operator) is addressed and proved. Moreover, the effectiveness of the proposed module is validated in experiments. The author also provided complete supplementary material to make the paper self-contained, especially the implementation details of pseudo torch code, which I found easy to follow and reproduce.

**Strength And Weaknesses:**

Pros:
- This paper provides a simplistic integration using VIN, which is easy to implement.
- The motivation to use VIN and the effectiveness of symmetric constraint is discussed and proved (only in 2D grid case).
- The experiment shows substantial improvements over baselines: VIN & GPPN.

Additionally, although I don't have experience in path & agent planning, I found this paper easy to understand.

Cons:
- Differences should be highlighted compared to E(2)-CNN, as they also proposed a steerable convolution with kernel constraints. Can their steerable convolution be applied to the same task with a little modification?
- Defining the generalized symmetric operation is challenging. Although the paper demonstrates the 2D grid case where the input signal (\mathbb{Z}) is rotated 90 deg, extending to 3D or higher dimension might be more difficult. For example, if the Euler angle represents the rotation in 3D, rotating a 3D signal with multiple 90 deg will suffer the gimbal lock.

**Summary Of The Paper:**

This paper proposes a novel steerable convolution for 2D path planning, where the convolution operation benefits from incorporating symmetric constraints to reduce overall search space and improve planning performance. Specifically, by manipulating the input signal in a symmetric setting (e.g. rotation and flipping for 2D space) and enforcing consensus outputs, the symmetric convolution can learn to reduce ambiguous prediction and therefore have better potential in both efficiency and accuracy.

**Summary Of The Review:**

I have several questions & concerns regarding the technique.
1. E(2)-CNN also proposed a transformation invariant representation yield by the steerable convolution. What is the difference compared to them? Can E(2)-CNN apply to the same task with slight modification?
2. Is the binary entropy loss applied to both raw and symmetric output?

---

> ### Author Response · Authors · 2022-11-16
> **Response to Reviewer QPW3**
>
> We appreciate the reviewer for the time and effort spent reviewing our work. We address the concerns through individual responses. We are happy to answer more concrete concerns the reviewer might have, such as "correctness (3: Some of the paper’s claims have minor issues. A few statements are not well-supported, or require small changes to be made correct)" and "technical novelty (2: The contributions are only marginally significant or novel.)".
>
> **Differences should be highlighted compared to E(2)-CNN, as they also proposed a steerable convolution with kernel constraints. Can their steerable convolution be applied to the same task with a little modification?**
>
> - We find this might be a misunderstanding of the equivariant network methods and our SymPlan methods.
> - We prove that value iteration in **2D** path planning is a steerable convolution, which can also be generalized to higher dimensions. Specifically, for the 2D discrete case, that has discrete rotation+reflection (D4) and translation symmetry.
>     - $E(2)$-steerable CNN can support discrete or continuous rotation+reflection+translation symmetry. Thus, ours is a case included in the $E(2)$-steerable CNN, since the 2D discrete grid has D4 symmetry (discrete rotation and reflection), which is a subgroup of the O(2) group (continuous rotation and reflection) in E(2) group. See Section E.2 for the details.
>     - So we show that if we want to add rotation and reflection to VIN, the resulting network is exactly an E(2)-steerable CNN (steerable convolution and other equivariant operations, in Equation 3). This also tells us how to design the representations (how to rotate the fields), such as in Figure 2 and Section D.1 (how to build Symmetric VIN).
> - Thus, our implementation directly uses the $E(2)$-steerable CNN (with discrete rotation and reflection equivariance) to replace regular planar 2D CNN (only translation equivariance), which follows our theory and requires only careful choice of representations. See the pseudocode in Section D.2.
>
> **Defining the generalized symmetric operation is challenging. Although the paper demonstrates the 2D grid case where the input signal (\mathbb{Z}) is rotated 90 deg, extending to 3D or higher dimension might be more difficult. For example, if the Euler angle represents the rotation in 3D, rotating a 3D signal with multiple 90 deg will suffer the gimbal lock.**
>
> - We agree that extending to a higher dimension is interesting while challenging. The goal of our work is to make the first step of integrating symmetry into (differentiable) planning, as suggested in the title.
> - There is a rich separate branch in math and physics on studying this (e.g., representing rotations or other transformations to 3D signals), representation theory.
>     - The reason we use 2D is to try to keep the representation part minimal, while providing an example pipeline on using equivariant networks for integrating symmetry. For 3D and other cases, the key is still equivariance, but they need more technical carefulness on the equivariant network side.
> - As for the example of 3D rotations, it has been intensively studied in representation theory. SO(3)-equivariant networks have also been studied in the equivariant network field, such as Cohen et al. (ICLR 2018, Spherical CNNs) and Geiger et al. (2022, e3nn: Euclidean Neural Networks).
>     - The reason SO(3)-equivariant networks do not suffer from gimbal lock is that they do not do pose prediction (output SO(3) elements) but only need equivariance to SO(3). Thus, SO(3) elements only appear in solving equivariant kernel constraints.
>     - Spherical CNN decomposes the signal by generalized Fourier transformation onto irreducible representations of $SO(3)$, which are the spherical harmonics. It then performs group convolutions operations (proven to be equivariant) on $S^2$ and $SO(3)$ and applies inverse Fourier transformation back to the spatial domain.

---

### Author Response · Authors · 2022-11-17
**Authors' general response**

We thank all reviewers for their thoughtful and detailed reviews! We address the concerns in individual responses.

We revised the paper for two things. As requested by the reviewer hMHM, we added the results of the SPL metric in Appendix Section B, so all other sections are shifted by a letter. We also change the color of citations from blue to brown, to make the new content highlighted in blue more noticeable.

---

### Decision · Program_Chairs · 2023-01-20

**Decision:**

Accept: poster

**Justification For Why Not Higher Score:**

The paper is interesting, but a better evaluation on realistic problems would have made it stronger.

**Justification For Why Not Lower Score:**

-

**Metareview: Summary, Strengths And Weaknesses:**

This paper introduces steerable convolutions into differentiable planning as an inductive bias. It received three expert reviews and quickly a consensus on acceptance emerged.

The reviewers appreciated the integration into VIN (differentiable planning) and the performance of the contribution, the theoretical formulation of equivariance. The weaknesses are the rather small toy problems, as it is unclear whether this contribution can generalize to realistic settings.

The AC agrees and estimates that these weaknesses are outweighted by the strong principled formulation of this problem.

**Note From Pc:**

if the above contains the word "oral" or "spotlight" please see: "oral" presentation means -> notable-top-5% and "spotlight" means -> notable-top-25%. As stated in our emails, we are disassociating presentation type from AC recommendations

**Summary Of Ac-Reviewer Meeting:**

There was no meeting.